

# Efficient ensemble generation for uncertain correlated parameters in atmospheric chemical models

Annika Vogel[1,2,3] and Hendrik Elbern[1,2]

[1]Institute for Energy and Climate Research - Troposphere (IEK-8), Forschungszentrum Jülich, Germany
[2]Rhenish Institute for Environmental Research at the University of Cologne, Germany
[3]Institute of Geophysics and Meteorology, University of Cologne, Germany

**Correspondence:** A. Vogel, av@eurad.uni-koeln.de

**Abstract.**

Atmospheric chemical forecasts highly rely on various model parameters, which are often insufficiently known, as emission rates and deposition velocities. However, a reliable estimation of resulting uncertainties by an ensemble of forecasts is impaired by the high-dimensionality of the system. This study presents a novel approach to efficiently perturb atmospheric-chemical model parameters according to their leading coupled uncertainties. The algorithm is based on the idea that the forecast model acts as a dynamical system inducing multi-variational correlations of model uncertainties. The specific algorithm presented in this study is designed for parameters which depend on local environmental conditions and consists of three major steps: (1) an efficient assessment of various sources of model uncertainties spanned by independent sensitivities, (2) an efficient extraction of leading coupled uncertainties using eigenmode decomposition, and (3) an efficient generation of perturbations for high-dimensional parameter fields by the Karhunen-Loéve expansion. Due to their perceived simulation challenge the method has been applied to biogenic emissions of five trace gases, considering state-dependent sensitivities to local atmospheric and terrestrial conditions. Rapidly decreasing eigenvalues state high spatial- and cross-correlations of regional biogenic emissions, which are represented by a low number of dominating components. Consequently, leading uncertainties can be covered by low number of perturbations enabling ensemble sizes of the order of 10 members. This demonstrates the suitability of the algorithm for efficient ensemble generation for high-dimensional atmospheric chemical parameters.

## 1 Introduction

Due to highly nonlinear properties of the atmosphere including its chemistry, forecast uncertainties vary significantly in space and time and among variables. During the last decades, increasing efforts have been put into estimating forecast uncertainties induced by different error sources. In this context, the method for generating an ensemble of forecasts is crucial as it determines the forecast probability distribution. While the represented details of the probability distribution increase with the number of realizations, the ensemble size of high-dimensional atmospheric systems is limited by computational resources (Leutbecher, 2019). Thus, the major challenge is the generation of ensembles which sufficiently sample the forecast uncertainty within manageable computational efforts. This renders ensemble forecasting one of the most challenging research areas in atmospheric modeling (e.g. Bauer et al., 2015; Buizza, 2019).





In numerical weather prediction (NWP), different ensemble methods have been developed in order to account for uncertainties of initial conditions and the forecast model formulation. First studies were motivated by the fact that initial conditions induce dominant uncertainties to NWP systems. Bred vectors (BV, Toth and Kalnay, 1993) or Singular vectors (SV, Buizza et al., 1993) are used to efficiently generate initial perturbations along the directions of the fastest growing errors in a linearized or nonlinear forecast model, respectively. Another approach estimates uncertainties of initial conditions by applying random

perturbations to observations (PO, Houtekamer et al., 1996) which are assimilated in the modeling system.

As errors in initial conditions cannot entirely explain forecast errors, two methods related to uncertainties within the NWP model have been developed. Firstly, the stochastic kinetic energy backscatter scheme (SKEBS, Shutts, 2005) accounts for uncertainties in the amount of energy which is backscattered from subgrid to resolved scales. The second group of methods focuses on uncertainties in model parameterizations, which rely on simplified assumptions about non-resolved processes. In

the stochastic parameter perturbation scheme (SPP, Houtekamer et al., 1996), selected parameters within individual parameterizations are multiplied with random numbers. In contrast, the stochastically perturbed parameterization tendencies scheme (SPPT, Buizza et al., 1999) considers uncertainties in the formulation of the parameterization schemes itself. Instead of perturbing individual parameters, total tendencies of state variables from all parameterizations are multiplied with appropriately scaled random numbers. Although perturbations are generated in a spatially and temporally correlated way, both, correlation

scales and standard deviations of the random numbers are predefined as fixed values (e.g., Leutbecher et al., 2017; Lock et al., 2019).

While different methods for ensemble generation are successfully applied to NWP, less approaches are available for chemistry transport modeling. As chemistry transport models (CTMs) include a large number of trace gases and aerosol compounds, the dimension of the system is even higher than in NWP (Zhang et al., 2012a). Among other implications, this high-

dimensionality amplifies the amount of uncertainties which differ significantly between individual chemical compounds (Emili et al., 2016). Besides using multi-model ensembles for estimation of forecast uncertainties (e.g., McKeen et al., 2007; Xian et al., 2019), there are only few attempts for ensemble generation within a single CTM. As CTMs are driven by meteorological forecasts, uncertainties in NWP are transferred to the chemical simulations. A comparably simple approach, which was used by Vautard et al. (2001) for the first time, employs an existing meteorological ensemble to drive the atmospheric chemical

forecasts. However, estimations of chemical uncertainties solely driven by NWP ensembles do not necessarily represent related uncertainties in CTMs. For example, Vogel and Elbern (2020) note that a global meteorological ensemble was not able to induce significant ensemble spread in surface-near forecasts of biogenic trace gases.

Multiple studies indicate that uncertainties of CTMs are mainly induced by uncertain model parameters – controlling emissions, chemical transformation and deposition processes – rather than initial conditions or meteorological forecasts (e.g., Elbern

et al., 2007; Bocquet et al., 2015). Consequently, former attempts aim to account for uncertainties in model parameters or other chemical input fields (for an overview see Zhang et al., 2012b, and references therein). However, perturbing parameter fields appears to suffer from the high-dimensionality of the system as independent perturbations of model parameters at each location and time remains impractical. Early studies like the one performed by Hanna et al. (1998) assume predefined uncertainties where perturbations are applied uniformly in space and time, ignoring any cross-correlations between parameters. This uniform





perturbation of model parameters with a fixed standard deviation is still applied to emissions in the context of ensemble data assimilation (e.g., Schutgens et al., 2010; Candiani et al., 2013).

However, constant perturbation of the whole parameter field does not allow for any spatial variation within the domain. More recently, limited spatial correlations are considered in uncertainty estimation by uniform perturbations within arbitrary sub-regions (Boynard et al., 2011; Emili et al., 2016) or isotropic decrease with fixed correlation length scales (Gaubert et al., 2014).

Although recent approaches allow a local treatment of correlations, they are not able to represent the spatio-temporal properties of the dynamical system. Already Hanna et al. (1998) propose that introducing state-dependent uncertainties as well as cross-correlations between parameters would provide a more realistic representation. The Karhunen-Loéve (KL) expansion provides an opportunity to account for such complex correlated uncertainties using eigenmode decomposition. While this approach is well established in engineering, it has rarely been applied in geophysical sciences. Goris and Elbern (2015) performed singular

vector decomposition to determine optimal placement of trace gas observation sites. Siripatana et al. (2018) used the KL expansion for dimension reduction in an idealized oceanographic ensemble data assimilation setup. To the knowledge of the authors, the KL expansion has not been applied in atmospheric chemistry ensemble modeling.

In order to address this issue, this study introduces a novel approach for optimized state-dependent parameter perturbation in atmospheric chemical models. The approach is based on the idea that the dynamical system induces multi-variational cor-

relations of model states and uncertainties. In particular, the algorithm aims to provide (1) an efficient assessment of various sources of uncertainties, (2) an efficient extraction of leading coupled uncertainties, and (3) an efficient generation of perturbations for high-dimensional parameter fields. The specific algorithm presented in Sect. 2 is designed for model parameters, which depend on model arguments like model inputs and configurations. Representative performance results are presented in Sect. 3 for biogenic emissions representing a highly uncertain, yet correlated set of parameters. Finally, Sect. 4 concludes this

study by discussing benefits and limitations of the presented approach.

## 2  Algorithm

This section provides the theoretical description of the ensemble generation algorithm with respect to correlated model parameters. Making use of the Karhunen-Loéve (KL) expansion, the algorithm is denoted as *KL ensemble algorithm* thereafter. It is based on the fact that the forecast model acts as a dynamical system forcing spatial and multi-variational couplings of the atmo-

spheric state. Thus, information on the size and coupling of forecast uncertainties can be extracted from differently configured model simulations. The explicit algorithm presented here focuses on state-dependent model parameters which depend on the specific model setup. Generally, atmospheric models are sensitive to their specific simulation setup including a large variety of model inputs and configurations – henceforth denoted as model arguments. These arguments comprise a heterogeneous set including initial conditions, external input fields and the formulation of model parameterizations. For state-dependent parame-

ters considered here, their sensitivities to the model setup are assumed to induce dominant uncertainties. Thus, the problem of estimating multi-variational uncertainties is transferred to sensitivities to various model setups.



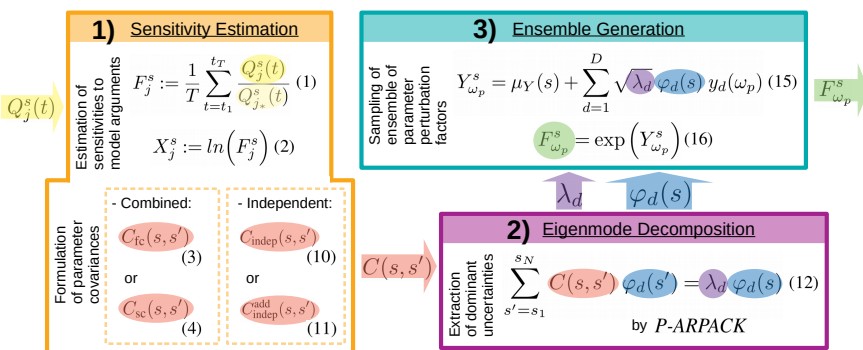

**Figure 1.** Simplified schematic overview of the KL ensemble algorithm. Equation numbers refer to the equations in Sect. 2. Colored arrows indicate input and output of the algorithm as well as transfer of selected fields between the single steps.

The algorithm consists of three major steps which are described in the following: the estimation of sensitivities to the model setup including the assumption of independent sensitivities (Sect. 2.1), the extraction of leading coupled parameter uncertainties using a highly-parallelized eigenmode decomposition software (Sect. 2.2), and the ensemble generation by sampling perturbations from leading eigenmodes with the Karhunen-Loéve expansion (Sect. 2.3). A graphical overview of the major steps composing the algorithm is given in Fig. 1.

## 2.1 Sensitivity Estimation

As a first step, essential uncertainties of the model parameters are formulated as multi-variational covariance matrix $C \in \mathcal{R}^{N \times N}$, where $N$ is the dimension of the problem i.e. the total dimension of the set of considered model parameters. Generally, the covariances may be determined from any kinds of uncertainties like statistical model errors derived from operational forecasts. Because this study focuses on state-dependent parameters, essential uncertainties are estimated from sensitivities of those parameters to different model arguments. These state-dependent sensitivities are realized as temporally-averaged sensitivity factors with respect to a selected reference. The temporal averaging makes the sensitivities being representative for a sufficient time interval for ensemble simulation.

For the formulation of the algorithm, each *model argument* $i \in [1, I]$ is interpreted as arbitrary parameter with $R_i$ different *implementations* $r_i \in [1, R_i]$. The implementations represent different available options for each model argument including initial fields, sources of input information or selections of model parameterizations. Let $Q_j^s(t)$ be the $j$-th forecast of a *model parameter* at discrete time $t \in [t_1, t_T]$ and position $s \in S$. Here, $s \in S$ denotes the *position* in the field of considered parameters at all grid points, which is specified by the *index set* $S = \{s_1, s_2, ..., s_N\}$ (analogous to Ch. 4.2 in Xiu, 2010). Note that each $j \in [0, J]$ represents one *model setup* $\{r_1, r_2, \ldots, r_I\}$ as specific combination of implementations $r_i$ of each model argument $i \in [1, I]$.

From a stochastic point of view, the model parameters $Q_j^s(t)$ can bee seen as induced pseudo-random variables. The pseudo-randomness reflects the fact that these variables are treated to be random, but are in fact controlled by the model setup. In





this regard, the selection of a specific implementation $r_i$ of a model parameter $i$ is a deterministic modification interpreted
as pseudo-stochastic realization. Then, the full set of available options $R_i$ of model argument $i$ is a sample subspace of the
respective argument space. Being aware of this stochastic interpretation, the algorithm is formulated using the application-
oriented notation introduced above. An overview over the notation used in this section including the definition of variables is
given in Appendix A.

The formulation of state-dependent sensitivities is based on Elbern et al. (2007) who demonstrated the suitability of am-
plification factors in the context of 4D-variational optimization of emissions. Let $j_*$ be a *reference model setup* (here $j_* := 1$)
representing the selected *reference implementation* $r_{i*} = 1$ of each model argument $i \in [1, I]$. Then, the model parameter $Q_j^s(t)$
of model setup $j$ at time $t$ and position $s$ is divided by its corresponding value from the reference configuration $Q_{j_*}^s(t)$. Thus,
the *sensitivity factor* $F_j^s$ is defined as temporal average of those over the time interval $[t_1, t_T]$:

$$F_j^s := \frac{1}{T} \sum_{t=t_1}^{t_T} \frac{Q_j^s(t)}{Q_{j_*}^s(t)} \qquad \forall \, j \in [1, J], \, s \in S \,. \tag{1}$$

Note that the sensitivity factors of the reference setup are one by definition: $F_{j_*}^s = 1 \quad, \forall \, s \in S$.

Analogous to emission factors in Elbern et al. (2007), sensitivity factors are assumed to be lognormally distributed. Thus,
the sensitivity factors are substituted to normally distributed *sensitivities* $X_j^s$ in order to simplify their further treatment:

$$X_j^s := \ln\left(F_j^s\right) \qquad \forall \, j \in [1, J], \, s \in S \,. \tag{2}$$

Given the formulation of sensitivities, two different methods for covariance construction are presented in the following.

### 2.1.1 Combined Sensitivities

Generally, the covariance matrix should represent the complete set of essential uncertainties of the model parameters. Focusing
on sensitivities to the model setup, essential uncertainties can be sampled using different implementations $r_i$ of multiple model
arguments $i \in [1, I]$. Here, $I$ is the total number of all model input and configuration options considered in the sensitivity anal-
ysis. Ideally, the covariance matrix is calculated from the sensitivities to all possible combinations of various implementations
of each option. Given $R_i$ implementations $r_i \in [1, R_i]$ of each model argument $i$, the total number of all combined sensitivities
$X_j^s$ with $j \in [1, J]$ is $J_{\mathrm{fc}} := J = \prod_{i=1}^{I} R_i$. With this, the *full combined covariance* between the sensitivities at positions $s, s' \in S$
is given by

$$C_{\mathrm{fc}}(s, s') := \frac{1}{J_{\mathrm{fc}} - 1} \sum_{j=1}^{J_{\mathrm{fc}}} \left( \left( X_j^s - \mu_{\mathrm{fc}}(s) \right) \cdot \left( X_j^{s'} - \mu_{\mathrm{fc}}(s') \right) \right) \quad, \tag{3a}$$

$$\text{where} \quad \mu_{\mathrm{fc}}(s/s') := \frac{1}{J_{\mathrm{fc}}} \sum_{j=1}^{J_{\mathrm{fc}}} X_j^{s/s'} \tag{3b}$$

is the mean value of full combined sensitivities at position $s$ and $s'$, respectively.



For atmospheric model parameters, each sensitivity $X_j^s$ requires its own forecast simulation. Thus, the full calculation of all combined sensitivities becomes computationally demanding even for a low number of implementations of a few model arguments. For example, considering six model arguments ($I = 6$) with two implementations ($r_i = 2, \forall\, i = [1, I]$) each, would already require $J_{\mathrm{fc}} = J = 2^6 = 64$ model executions prior to the ensemble generation. One method to overcome this issue is to randomly sample a subset of $J_{\mathrm{sc}} < J_{\mathrm{fc}}$ combined sensitivities and approximate the *sampled combined mean and covariance* by

$$C_{\mathrm{sc}}(s, s') := \frac{1}{J_{\mathrm{sc}} - 1} \sum_{j=1}^{J_{\mathrm{sc}}} \left( \left( X_j^s - \mu_{\mathrm{sc}}(s) \right) \cdot \left( X_j^{s'} - \mu_{\mathrm{sc}}(s') \right) \right) \quad , \tag{4a}$$

$$\text{with} \quad \mu_{\mathrm{sc}}(s/s') := \frac{1}{J_{\mathrm{sc}}} \sum_{j=1}^{J_{\mathrm{sc}}} X_j^{s/s'} \quad . \tag{4b}$$

However, a reasonable covariance estimation by random sampling still requires a large number of combined sensitivities which would be in contradiction to the idea of an efficient ensemble generation approach. Thus, a different method is introduced below, which is based on the assumption of independent sensitivities.

### 2.1.2 Independent Sensitivities

As this study aims for an computationally efficient algorithm focusing on leading uncertainties, the computational efforts required for the estimation of sensitivities are critical. Thus, a new method for efficient covariance construction is introduced, which reduces the number of required model executions prior to ensemble generation significantly. Instead of using a randomly sampled subset of combined sensitivities, the method uses only sensitivities with respect to single model arguments. By assuming tangent linearity of sensitivities in the limits of imposed perturbations, these single sensitivities are extrapolated to approximate the full set of combined sensitivities. The assumption of tangent linearity equals mutual independence of the model arguments and thus every combined sensitivity factor $F_j^s$ with arguments $\{r_1, r_2, \ldots, r_I\}$ can be decomposed into a set of *independent sensitivity factors* $f_i^s$ to each single argument $r_i$ with $i \in [1, I]$

$$F_j^s = F_{\{r_1, r_2, \cdots, r_I\}}^s = \prod_{i=1}^{I} f_i^s(r_i) \qquad \forall\, j \in [1, J]\,, s \in S \quad . \tag{5}$$

Here, the independent sensitivity factors $f_i^s$ are defined analogous to Eq. (1) using the model forecast $q_i^s(r_i, t) := Q_{\{r_{1*}, r_{2*}, \ldots, r_{i-1*}, r_i, r_{i+1*}, \ldots, r_{I*}\}}^s(t)$ were only one argument $r_i$ differs from the reference setup:

$$f_i^s(r_i) := \frac{1}{T} \sum_{t=t_1}^{t_T} \frac{q_i^s(r_i, t)}{Q_{j*}^s(t)} \qquad \forall\, i \in [1, I],\, s \in S\,. \tag{6}$$

Further, with Eq. (2), every combined sensitivity $X_j^s$ of implementation $j$ at position $s$ is given by

$$X_j^s = X_{\{r_1, r_2, \cdots, r_I\}}^s = ln\left( F_{\{r_1, r_2, \cdots, r_I\}}^s \right) \overset{(5)}{=} \sum_{i=1}^{I} ln\left( f_i^s(r_i) \right) = \sum_{i=1}^{I} x_i^s(r_i) \quad , \tag{7}$$





were $x_i^s(r_i) := ln\left(f_i^s(r_i)\right)$ is the *independent sensitivity* referring to a single modified model argument $i \in [1, I]$ with implementation $r_i$. Note that the independent sensitivity factors $f_i^s(r_i)$ equal one when the implementation $r_i$ of model argument $i$

170 equals the reference implementation $r_{i*}$ (analogous to Eq. (1) ). Consequently, the independent sensitivities $x_i^s(r_i)$ vanish in Eq. (7) for all $i$ with $r_i = r_{i*}$ and each combined sensitivity is given by the sum of independent sensitivities to those arguments, which differ from the reference setup

$$X_j^s = X_{\{r_1, r_2, \cdots, r_I\}}^s = \sum_{i=1 \,|\, r_i \neq r_{i*}}^{I} x_i^s(r_i) \qquad \forall\, j \in [1, J]\,, s \in S \quad , \tag{8}$$

In other words, the assumption of independence implies that the set of all combined sensitivities $\left\{X_j^s\right\}_{j \in [1, J]}$ lies within a

175 subspace which is spanned by the set of independent sensitivities $\left\{x_i^s(r_i)\right\}_{\substack{r_i \in [1, R_i] \\ i \in [1, I]}}$. Then, the full set of combined sensitivities can be approximated by the subset of independent sensitivities following Eq. (8). This reduces the number of required forecasts from $J_{\mathrm{fc}} = \prod_{i=1}^{I} R_i$ to:

$$J_{\mathrm{indep}} = 1 + \sum_{i=1}^{I} (R_i - 1) \quad , \tag{9}$$

with $J_{\mathrm{indep}} \ll J_{\mathrm{fc}} = J$. Therewith, the independent method requires a significantly reduced number of simulations, one for the

180 reference setup ($r_{i*} = 1 \,\forall i$) and one for each other implementation $r_i \in [2, R_i]$ of each argument $i \in [1, I]$.

Rather than approximating all combined sensitivities from independent sensitivities explicitly, the effects on mean sensitivities and covariances are derived in its general from in Appendix B. In the KL ensemble algorithm, the mean values $\mu_{\mathrm{indep}}(s)$ and covariances $C_{\mathrm{indep}}(s, s')$ of the sensitivities are directly calculated from the set of independent sensitivities by

$$\mu_{\mathrm{indep}}(s) \overset{(B3)}{=} \sum_{i=1}^{I} \left( \frac{1}{R_i} \cdot \sum_{r_i=1}^{R_i} x_i^s(r_i) \right) \overset{(8)}{=} \sum_{i=1}^{I} \left( \frac{1}{R_i} \cdot \sum_{r_i=1 \,|\, r_i \neq r_{i*}}^{R_i} x_i^s(r_i) \right) \tag{10a}$$

185 $$C_{\mathrm{indep}}(s, s') \overset{(B4)}{=} \frac{J}{J-1} \sum_{i=1}^{I} \left[ \left( \frac{1}{R_i} - \frac{1}{(R_i)^2} \right) \cdot \sum_{r_i=1}^{R_i} \left( x_i^s(r_i) \cdot x_i^{s'}(r_i) \right) \right]$$

$$\overset{(8)}{=} \frac{J}{J-1} \sum_{i=1}^{I} \left[ \left( \frac{1}{R_i} - \frac{1}{(R_i)^2} \right) \cdot \sum_{r_i=1 \,|\, r_i \neq r_{i*}}^{R_i} \left( x_i^s(r_i) \cdot x_i^{s'}(r_i) \right) \right] \quad . \tag{10b}$$

Note that the assumption of independence does not imply orthogonality between the input sensitivities. While the equations are exact under the given assumption, it might not be a sufficient approximation for many atmospheric processes.

The method of independent sensitivities allows the inclusion of additional uncertainties in a straightforward way. These

190 additional uncertainties may originate form any other error source not represented as model arguments. For example, this could be a known uncertainty in the formulation of the model itself. If such an *additional uncertainty* is given (e.g. from statistical evaluation), it can be included as additional sensitivity $x_{\mathrm{add}}^s$ with $R_{\mathrm{add}} = 2$. Based on Eq. (10), the independent mean





and covariance including additional uncertainties are

$$
\mu_{\text{indep}}^{\text{add}}(s) = \sum_{i=1}^{I} \left( \frac{1}{R_i} \cdot \sum_{r_i=1 \mid r_i \neq r_{i*}}^{R_i} x_i^s(r_i) \right) + \frac{1}{R_{\text{add}}} \cdot x_{add}^s \tag{11a}
$$

$$
C_{\text{indep}}^{\text{add}}(s,s') = \frac{J}{J-1} \left( \sum_{i=1}^{I} \left[ \left( \frac{1}{R_i} - \frac{1}{(R_i)^2} \right) \cdot \sum_{r_i=1 \mid r_i \neq r_{i*}}^{R_i} \left( x_i^s(r_i) \cdot x_i^{s'}(r_i) \right) \right] + \frac{1}{(R_{\text{add}})^2} \cdot \left( x_{\text{add}}^s \cdot x_{\text{add}}^{s'} \right) \right) \quad . \tag{11b}
$$

If the direction of the additional uncertainty is unknown (*unsigned additional uncertatinty*), the original definition of the mean values for independent sensitivities as given in Eq. (10a) is used instead of Eq. (11a). This ensures no impact of the additional uncertainty to the mean values of the parameters.

## 2.2 Eigenmode Decomposition

Once the multi-variational covariances are formulated, dominating directions of uncertainties are extracted as second step. This extraction is realized by an eigenmode decomposition of the covariance matrix

$$
\sum_{s'=s_1}^{s_N} C(s,s') \, \varphi_d(s') = \lambda_d \, \varphi_d(s) \quad , \tag{12}
$$

with $\lambda_d$ the d-th eigenvalue and $\varphi_d(s/s')$ the $s/s'$-th element of the corresponding eigenvector $\varphi_d \in \mathcal{R}^N$ for all $d \in [1, N]$. As the presented ensemble approach focuses on dominant uncertainties, the $D$ largest eigenvalues and corresponding eigenvectors are required $\lambda_1 \geq \lambda_2 \geq \cdots \geq \lambda_D$ with $D < N$. Here, the first eigenvalues represent the size of the most dominant uncertainties and the corresponding eigenvectors their directions. Due to the high-dimensionality of atmospheric models, the covariance matrix may easily be of the order of $10^{10}$ elements. This inhibits explicit storage of the matrix and makes the computation of the eigenproblem Eq. (12) very costly even for a low number of required eigenmodes ($D \ll N$). Therefore, a highly efficient software is required which is suitable for high-dimensional systems.

The ARPACK (ARnoldi PACKage, Lehoucq et al., 1997) package is a flexible tool for numerical eigen and singular value decomposition. It is explicitly developed for large-scale problems and includes a set of specific algorithms for different types of matrices. The ARPACK software uses a reverse communication interface were the matrix needs to be given as operator acting on a given vector. APRACK makes use of the Implicitly Restarted Arndoldi Method (IRAM, Sorensen, 1997) which is based on the implicitly shifted QR-algorithm. As a covariance matrix is quadratic, symmetric and positive definite by construction, the IRAM method reduces to the Implicitly Restarted Lanczos Method (IRLM). In this study, the parallel version P-ARPACK is used for the eigenmode decomposition, which balances the workload of processors and reduces the computation time. For a detailed description of the ARPACK software package see Lehoucq et al. (1997).

## 2.3 Ensemble Generation

The final step is the generation of an ensemble of perturbations based on the leading eigenmodes of parameter uncertainties. This step makes use of the Karhunen-Loéve expansion – denoted as *KL expansion* thereafter – named after Karhunen (1947)





and Loéve (1948). The following description is adopted from the notations of Schwab and Todor (2006) and Xiu (2010), to which the reader is referred for more details. In its discrete form, the KL expansion describes the s-th element of a stochastic process $Y_{\omega_p}^s$ of dimension $N$ as linear combination of orthogonal components

$$Y_{\omega_p}^s = \mu(s) + \sum_{d=1}^{N} \psi_d(s)\, y_d(\omega_p) \quad , \tag{13}$$

with $\mu(s)$ denoting the mean value of the stochastic process and $\omega_p$ its p-th random realization. Here, the deterministic fields $\psi_d(s)$ are given by the eigenvalues $\lambda_d$ and eigenvectors $\varphi_d$ of the covariances of the stochastic process

$$\psi_d(s) := \sqrt{\lambda_d}\, \varphi_d(s) \tag{14}$$

In this notation, the stochastic coefficients $y_d(\omega_p)$ are independent random numbers with zero mean and unit standard deviation.

In the context of ensemble generation, the stochastic process is a set of *perturbations* $\left\{ Y_{\omega_p}^s \right\}_{s \in S}$ whose essential uncertain-
ties are formulated as covariance matrix of the sensitivities as defined in Sect. 2.1. Thus, the eigenvalues $\lambda_d$ and corresponding normalized eigenvectors $\varphi_d(s)$ are provided by the eigenmode decomposition in Sect. 2.2. Normally distributed sensitivities $X_j^s$ can be realized by centered and normally distributed stochastic coefficients $y_d(\omega_p)$.

Using the KL expansion for ensemble generation, multivariate covariances induce coupled perturbations of the set of considered parameters. The higher the correlations of the sensitivities, the faster is the decrease of the eigenmodes and the more are
the perturbations determined by a few leading orthogonal components. Truncating Eq. (13) at $D < N$, the resulting KL approximation provides an optimal approximation of the stochastic process in the least-square sense (Schwab and Todor, 2006). For ensemble generation, a set of $D$ stochastic coefficients $y_d(\omega_p)$ is randomly sampled for each ensemble member $p$ from a normal distribution with zero mean and unit standard deviation. Given the set of leading eigenvalues $\left\{ \lambda_d \right\}_{d \in [1,D]}$ and corresponding normalized eigenvectors $\left\{ \phi_d(s) \right\}_{\substack{s \in S \\ d \in [1,D]}}$, the perturbation of ensemble member $p \in [1,P]$ at position $s \in S$ is sampled from

$$Y_{\omega_p}^s = \mu_Y(s) + \sum_{d=1}^{D} \sqrt{\lambda_d}\, \varphi_d(s)\, y_d(\omega_p) \qquad \forall\, \omega_p \in [\omega_1, \omega_P],\ s \in S \,. \tag{15}$$

Finally, the ensemble of perturbations is resubstituted to a set of *perturbation factors* $\left\{ F_{\omega_p}^s \right\}_{s \in S}$ analogous to the definition of sensitivities in Eq. (2)

$$F_{\omega_p}^s = \exp\left( Y_{\omega_p}^s \right) \qquad \forall\, \omega_p \in [\omega_1, \omega_P],\ s \in S \,. \tag{16}$$

These perturbation factors will then be applied to the model parameters in the ensemble forecast.

## 3 Application to Biogenic Emissions

The KL ensemble algorithm was implemented and integrated into the chemical data assimilation system EURAD-IM (*EURopean Air pollution Dispersion - Inverse Model*). EURAD-IM combines a state-of-the-art chemistry transport model (CTM)





with 4-dimensional variational data assimilation (Elbern et al., 2007). Based on meteorological fields precalculated by WRF-ARW (*Weather Research and Forecasting - Advanced Research WRF*, Skamarock et al., 2008), the Eulerian CTM performs

forecasts of about 100 gas phase and aerosol compounds up to lower stratospheric levels. In addition to advection and diffusion processes, modifications due to chemical conversions are considered by the RACM-MIM chemical mechanism (*Regional Atmospheric Chemistry Mechanism - Mainz Isoprene Mechanism*, Pöschl et al., 2000; Geiger et al., 2003). Emissions from anthropogenic and biogenic sources as well as dry and wet deposition act as chemical sources and sinks, respectively. In this study, the EURAD-IM system provides forecasts of sensitivities to various model arguments used for covariance construction.

The concept of emission factors used for emission rate optimization in EURAD-IM was adapted in the KL ensemble approach.

The KL ensemble algorithm is applied to biogenic emissions which are known to be subject to large uncertainties. The MEGAN 2.1 model developed by Guenther et al. (2012) calculates biogenic emissions of various compounds in EURAD-IM as function of atmospheric and terrestrial conditions including radiation, air temperature, leaf area index, and soil moisture. In this study, biogenic emissions of five dominant volatile organic compounds (VOC) are perturbed: isoprene, limonene, alpha-

pinene, ethene, and aldehydes. Note that biogenic aldehyde emissions from MEGAN 2.1 represent the total emission from acetaldehyde and a set of higher aldehydes which are not treated individually (see Guenther et al., 2012, for further details). The set of five biogenic emission fields is selected in order to investigate the joint perturbation of highly uncertain, yet correlated parameters. The following evaluation of the results focuses on emissions of isoprene, which is the most abundantly emitted biogenic trace gas.

The sensitivities used for covariance construction are taken from a case study covering the Po valley in northern Italy on 12.07.2012. As shown in a preceding study by Vogel and Elbern (2020), local emissions of biogenic volatile organic compounds (BVOCs) in this case study are highly sensitive to various model arguments. At the same time, these sensitivities are found to be almost species invariant and show little variation on an hourly timescale which allows for a generalized formulation of perturbations. Providing an appropriate test case, the sensitivities used in this study are based on the results of Vogel and Elbern

(2020) simulated by EURAD-IM.

Specifically, emission sensitivity factors $F_j^s$ are calculated from hourly biogenic emissions divided by the corresponding reference emissions and averaged over the period on 12.07.2012 from 00 UTC until 10 UTC according to Eq. (1). Here, minimum emissions of $1.0 \cdot 10^{-3} \frac{kg}{km^2\,h}$ are defined and sensitivity factors are limited by 0.1 and 10.0 in order to avoid unrealistic perturbations in regions of low emissions. As biogenic emissions are restricted to terrestrial vegetation, only land surface gridboxes

are used, which reduces the total dimension of the problem by about 27 % compared to all surface gridboxes.

Based on this, a representative example of the ensemble generation algorithm are presented using (i) combined sensitivities and (ii) independent sensitivities including additional uncertainties. These two methods are selected to show potentials and limitations of the algorithm.

### 3.1   Sensitivity Estimation

Six different model arguments are considered in this study ($I = 6$): land use information, global meteorology, land surface model, boundary layer & surface layer parameterizations, cloud microphysics parameterization, an short & longwave radiation





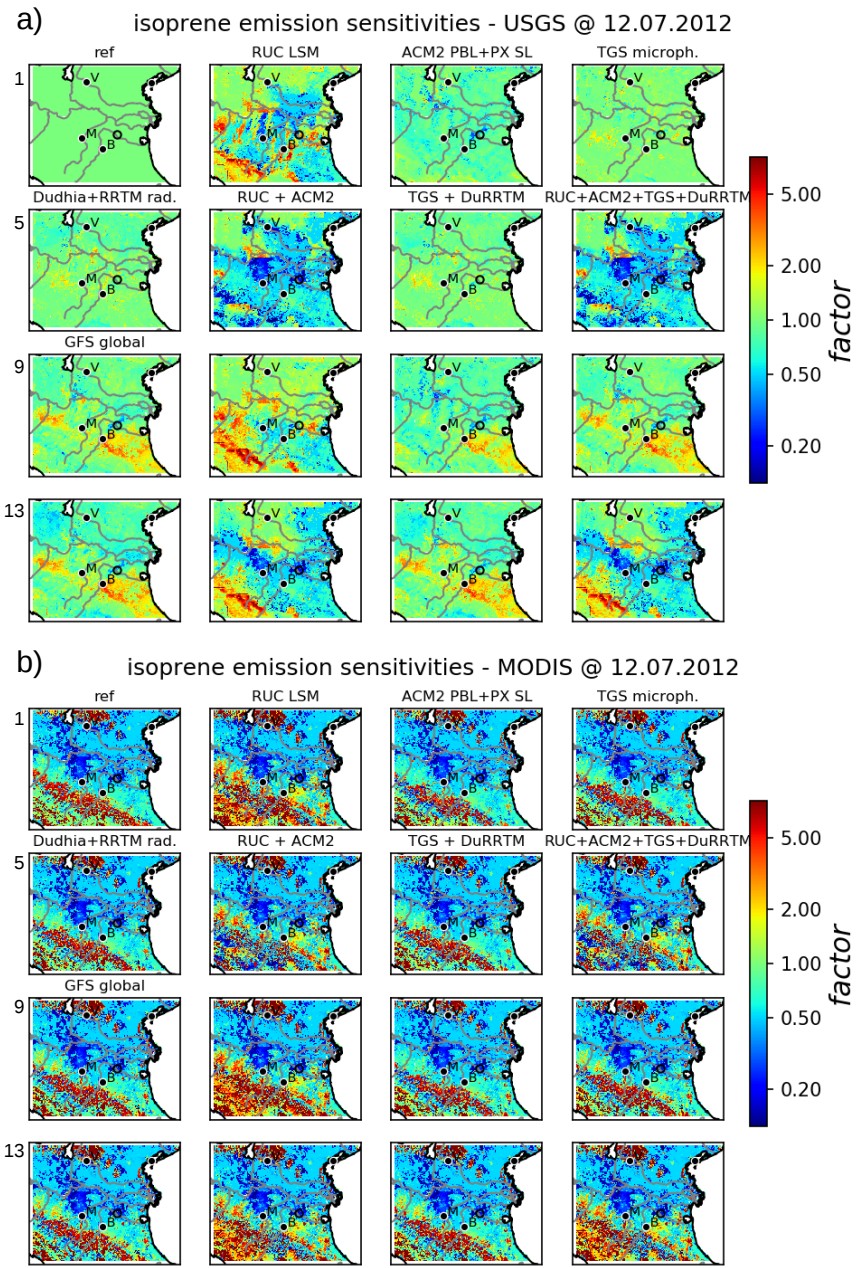

**Figure 2.** Combined sensitivities of isoprene splitted for USGS (a) and MODIS (b) land use, as simulated by EURAD-IM. The sensitivities are given as factors w.r.t reference emissions and ordered according to Tab. C1 from left to right and from top to bottom. Some major cities (Verona, Bologna, Modena) are indicated by their initial letters.



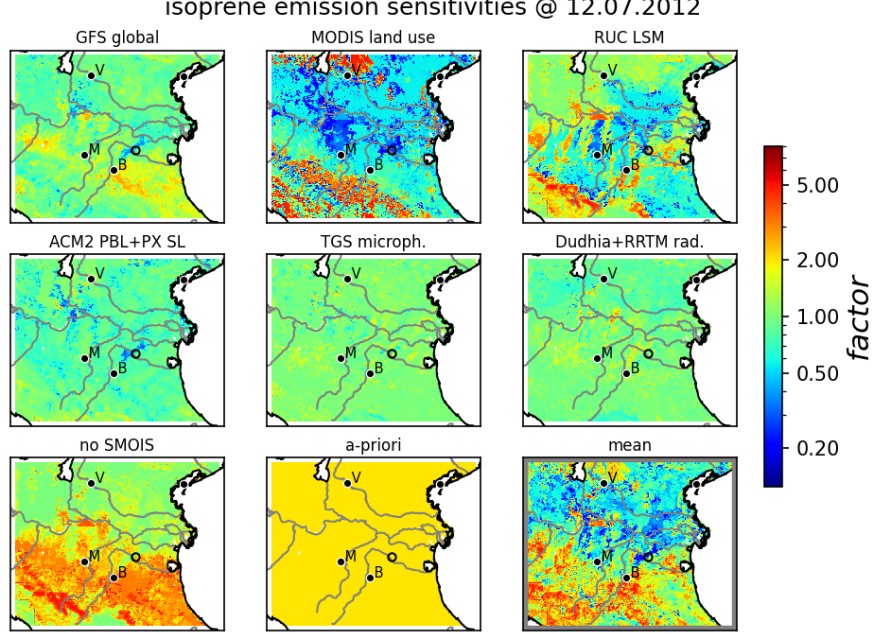

**Figure 3.** Independent mean (lower right panel) and sensitivities of isoprene including additional uncertainties due to drought response ('no SMOIS') and the emission model ('a-priori') of biogenic emissions simulated by EURAD-IM. Plotting conventions as in Fig. 2.

parameterizations (see Vogel and Elbern, 2020, for further details). For each argument $i \in [1,6]$, two different implementations are selected ($R_i = 2, \ \forall \ i$), the first of which is defined as reference implementation ($r_{i*} = 1, \ \forall \ i$). Figure 2 shows 32 combined sensitivity factors of biogenic emissions calculated from EURAD-IM using Eq. (1) with different combinations of model

configurations as listed in Tab. C1. For computational reasons, the subset of $J_{sc} = 32$ combined sensitivities is sampled from a total number of $J = 2^6 = 64$ possible combinations. Thus, mean emission sensitivities and covariances are calculated from Eq. (4).

According to the definition in Eq. (1), sensitivity factors of the reference run (panel 1 of Fig. 2a) are equal to 1 by definition. The sensitivity factors are dominated by effects of land use information shown in Fig. 2b, inducing reduced or increased

emissions in the mountains and the Po valley, respectively. As discussed in Vogel and Elbern (2020), these large sensitivities in biogenic emissions are caused by different fractions of broadleaf trees in USGS land use information and MODIS data. Significant effects are also found with respect to global meteorology, land surface model (LSM) and boundary layer schemes. Here, weak nonlinear effects appear when RUC LSM is combined with the ACM2 boundary layer scheme or GFS global meteorology (panel 6 and 10 of Fig. 2a).

In contrast to the large amount of calculations required for combined sensitivities, the method of independent sensitivities is additionally investigated. As described in Sect. 2.1.2, only sensitivities resulting from the change of a single model argument



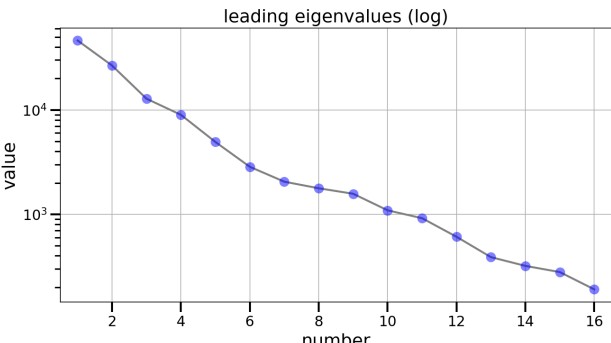

**Figure 4.** Leading eigenvalues of biogenic emissions for combined sensitivities (blue dots). Eigenvalues are plotted on a logarithmic scale.

are required for this method. This allows for an additional consideration of two uncertainties related to the emission model. Firstly, the highly variable response of biogenic emissions to soil dryness is added to the set of independent sensitivities. This sensitivity is defined as the change of emissions when the drought response used in MEGAN2.1 is excluded (compare Vogel

and Elbern, 2020). Secondly, Guenther et al. (2012) indicate an uncertainty of the emissions model itself of $200\,\%$, which is included as unsigned additional uncertainty (denoted as *a-priori uncertainty*) with a constant factor of two for all locations and trace gases. Note that an unsigned additional uncertatinty does only apply to the covariances given by Eq. (11b) and does not affect the calculation of the independent mean given by Eq. (10a). The formulation of a constant factor induces a simple assumption representing perfectly correlated errors in this case. But it is assumed to be sufficient to show the effect of including

unsigned additional uncertainties in the algorithm.

Figure 3 shows the independent mean factors and sensitivities for isoprene emissions. Note that independent sensitivities are formulated relative to the independent mean, which are both limited by $0.2$ and $5$ to be consistent with the configuration for combined sensitivities. Similar to the results of combined sensitivities, MODIS land use information and RUC land surface model produce significantly reduced isoprene emissions within the northern Po valley. The added sensitivity to drought

response points towards increased emissions in the southern part of the domain. Consequently, the independent mean is dominated by reduced emissions in the northern part and increased values in the southern part. The a-priori uncertainty of the emission model is represented by a constant factor of two and does not affect the independent mean by definition. The remaining independent sensitivities produce only minor deviations in biogenic emissions of all trace gases including isoprene.

### 3.2 Eigenmode Decomposition

Based on the respective formulation of combined or independent sensitivities, the leading eigenvalues and their associated eigenvectors of the covariance matrices are calculated. For combined sensitivities, the eigenvalues given in Fig. 4 show a logarithmic decrease of about one order of magnitude within the first five modes. This indicates that the major uncertainties of the emissions factors are determined by a few leading directions. In other words, the fast decrease of leading eigenvalues confirms a high correlation of biogenic emissions through the domain and between different gases. The contribution of these




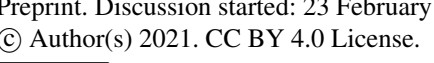

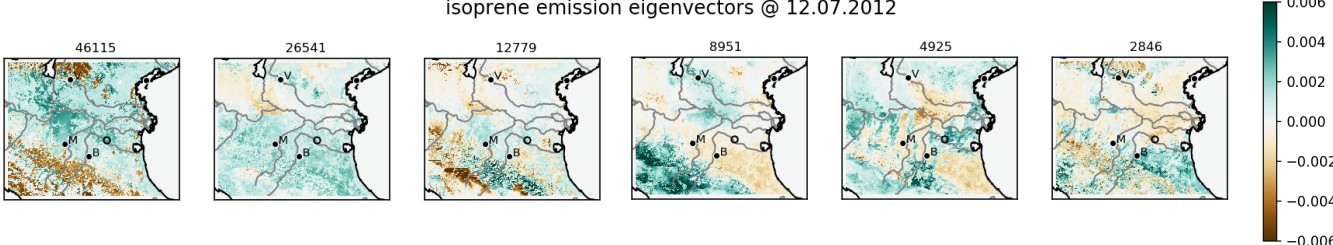

**Figure 5.** Isoprene-components of normalized leading eigenvectors for combined sensitivities. Corresponding eigenvalues are given above each eigenvector. Some major cities (Verona, Bologna, Modena) are indicated by their initial letters.

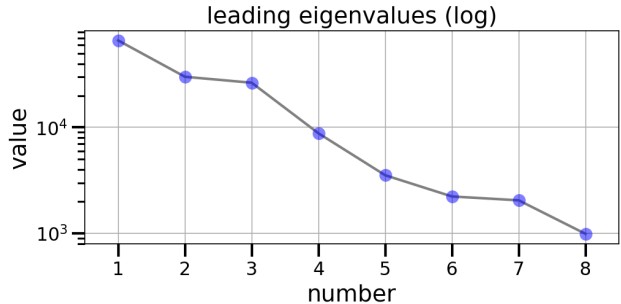

**Figure 6.** Leading eigenvalues of biogenic emissions for independent sensitivities including additional uncertainties. Plotting conventions as in Fig. 4.

leading eigenmodes to local emission factors for each trace gas is given by the corresponding eigenvectors shown in Fig. 5 for isoprene. According to shape and size of the first eigenmode, it is almost exclusively induced by the sensitivity to land use information which is invariant to the other sensitivities. The subsequent eigenmodes represent common patterns of the remaining sensitivities which are therefore treated together.

Figure 7 shows the complete leading eigenvectors for independent sensitivities with respect to all five biogenic trace gases.
As for combined sensitivities, the eigenmode decomposition extracts perpendicular components from the set of independent sensitivities. The first eigenmode represents common features of the a-priori uncertatinty and the other independent sensitivities. While the second eigenmode is closely related to the effects of drought response and the land surface model, the sensitivity to land use dominates the third eigenmode.

The consideration of additional uncertainties in the independent method does not allow for a direct comparison of individual
eigenmodes. The eigenvalues shown in Fig. 6 state a similar decrease of eigenvalues for independent sensitivities compared to the combined method. Highly similar size and decrease rate of leading eigenvalues indicate a reasonable representation of the leading uncertainties by the independent method. However, nonlinearities arising from combined changes in the land surface model with global meteorology or boundary layer schemes are not captured by the linear assumption of independent sensitivities. Besides that, highly similar signals in eigenvectors for different biogenic gases shown in Fig. 7 state a considerably

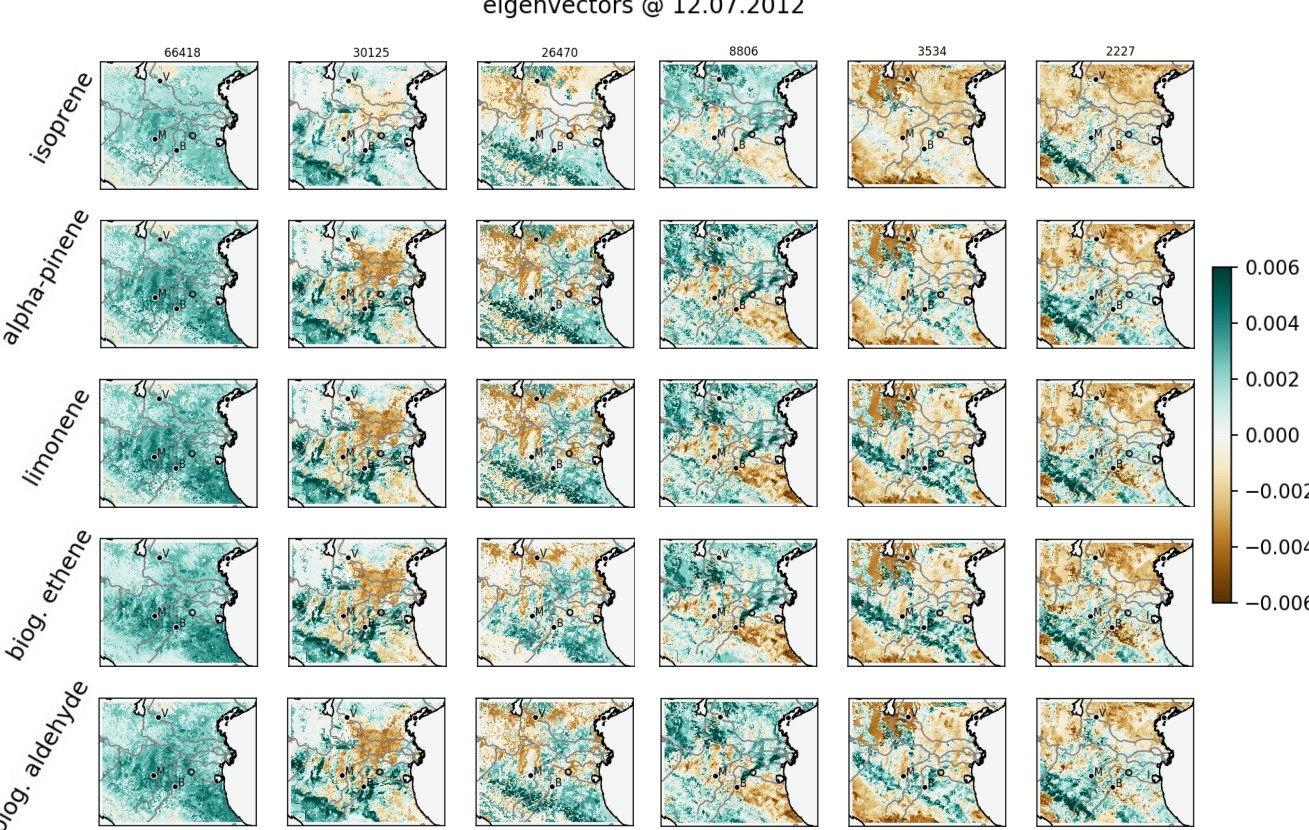

**Figure 7.** Normalized leading eigenvectors for independent sensitivities including additional uncertainties. Corresponding eigenvalues are given above each eigenvector visualized as column of fields of different biogenic gases. Plotting conventions as in Fig. 5.

large correlation between those. Yet, individual patterns of each biogenic gas are also represented by the leading eigenmodes for independent sensitivities. These patterns are also found for the eigenvectors of combined sensitivities (not shown) which confirms the suitability of the independent method with respect to multiple parameters.

### 3.3 Ensemble Generation

The different directions of the leading eigenmodes of the two methods prohibit a direct comparison of their perturbations. Due to the sufficient representation of dominant uncertainties and additional consideration of model-induced uncertainties, resulting perturbations are only shown for independent sensitivities.

Figure 8 shows eight realizations of perturbations in terms of emission factors for isoprene for independent sensitivities. The perturbation factors of all biogenic emissions are calculated by multiplying the independent mean factors of the sensitivities with different realizations of the KL expansion. As the KL expansion does not affect the mean values, ensemble mean emission factors remain similar to the one of the sensitivities (compare Fig. 3). Although differences in perturbation factors are large, this



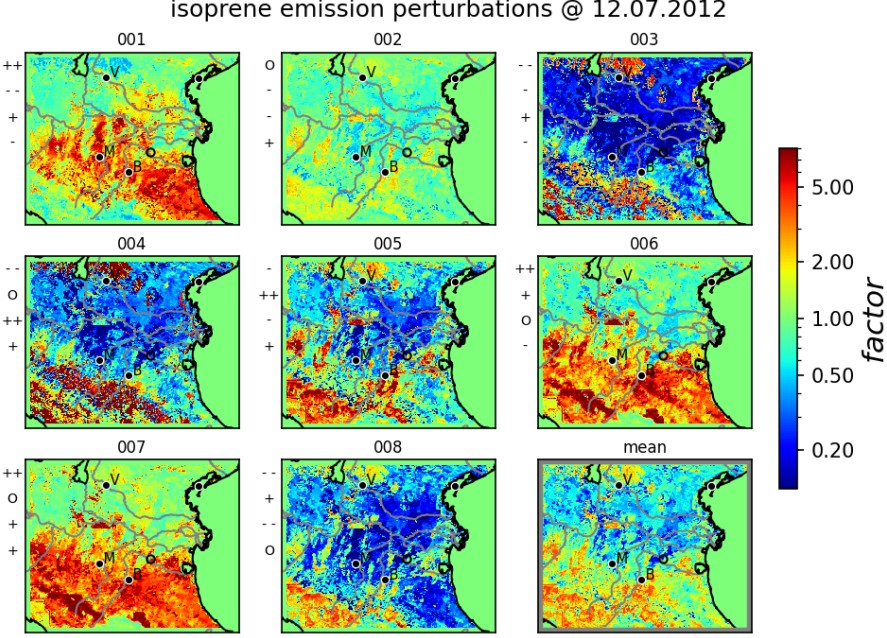

**Figure 8.** Perturbations of isoprene emissions for independent sensitivities including additional uncertainties given as factors w.r.t reference emissions. Random realizations of stochastic coefficients for the leading eigenmodes of each member are indicated left of each subplot $\Big($ '**++**': large positive value ($> 1.$), '**+**': small positive value ($0.1 < * < 1.$), 'O': very small absolute value ($-0.1 < * < 0.1$), '**–**': small negative value ($-1. < * < -0.1$), '**– –**': large negative value ($< -1.$) $\Big)$. The lower right subplot gives the ensemble mean factors. Some major cities (Verona, Bologna, Modena) are indicated by their initial letters.

suggests a reasonable sampling of the eight realizations. Concerning the individual members, the high number of significant uncertainties results in emission factors ranging up to more than one order of magnitude. Each realization is influenced by different combinations of the leading eigenmodes resulting in different perturbation patterns. While realization 001, 006 and 007 are dominated by a positive contribution of the first eigenmode, the effect of the second mode is clearly visible when comparing realization 001 and 006. Comparing realization 004 and 008, the most significant differences are induced by the third eigenmode. Due to the fast decrease of eigenmodes, comparably small contributions of the remaining modes remain invisible in the perturbation factors.

# 4 Discussion and Conclusions

This study introduces an optimized ensemble generation algorithm in which model parameters are efficiently perturbed according to their correlations. The approach is based on the fact that advection–diffusion–reaction equations including their forcing parameters act as dynamical systems which induces multi-variational correlations of model states and uncertainties. It applies





the Karhunen-Loéve expansion which approximates covariances of the model parameters by a limited set of leading eigen-modes. These modes represent the coupled leading uncertainties from which perturbations can be sampled efficiently. Based on this, stochastic sampling for ensemble generation is performed in an uncorrelated subspace spanned by the eigenmodes.

Generally, the presented algorithm is applicable to any set of model parameters in high-dimensional atmospheric systems, as long as their joint uncertainties remain in the linear regime. Through the reduction of the sampling space, it is shown that the stochastic dimension of the problem can be reduced significantly. This makes the algorithm suitable for efficient ensemble generation of high-dimensional atmospheric models, where the computational costs are a critical and limiting quantity.

In the Karhunen-Loéve (KL) ensemble algorithm, perturbations are created from covariances of the stochastic process.

This means that for large ensemble sizes the statistics of the perturbations converge towards their input values determined by the covariances. Consequently, the performance of the KL ensemble crucially relies on the formulation of the covariances. Uncertainties not considered in the formulation of the covariances cannot be captured by the KL ensemble. The major benefit of the approach lies in the optimal properties of the perturbations focusing on leading uncertainties, providing an optimal coverage of uncertainties even for low ensemble sizes. Although the greatest benefit is achieved for highly correlated parameters, the

algorithm enables the combination of major uncertainties even for uncorrelated parameters.

Focusing on model parameters which depend on local environmental conditions, state-dependent covariances are approximated from various related sensitivities. Generally, the covariances required for this approach can be defined in any way which is suitable to reflect the uncertainty of our knowledge. Covariance construction based on parameter sensitivities as presented in this study is just one among others. Potential deficiencies in the construction could be identified from a posteriori evaluation

of the full ensemble. This would allow for an ongoing adjustment of the approach depending on the specific application.

As simulations of all possible combinations of sensitivities are computationally demanding, independent sensitivities are introduced in this study. Assuming tangent-linearity, multiple combined sensitivities can be represented by a low number of independent sensitivities. Representative results indicate that the major properties of leading sensitivities are captured by independent sensitivities. At the same time, using only 7 instead of 32 pre-calculated sensitivities reduces the computational effort

of model simulations tremendously. Besides the reduction of computational resources, this method allows for the integration of different kinds of uncertainties in a convenient way. However in many cases, the assumption of independent sensitivities may not be a good approximation. The user has to decide if the computational benefit justifies the neglection of nonlinear effects.

Once the sensitivities are calculated, the computational effort required for the generation of Karhunen-Loéve (KL) perturbations is mainly consumed by the numerical solution of the eigenproblem. The highly efficient parallelization of the solution

of the eigenproblem by parallel-ARPACK renders the algorithm suitable for high-dimensional systems. Table 1 summarizes the computation time and relevant properties for selected setups. By definition, the computing time is proportional to the size of the covariance matrix, which increases quadratically with the dimension of the considered model parameters. Note that the computing time is given as wall clock time for calculating the perturbations. Due to the parallelization of the computation, the total CPU time scales linearly with the number of cores used. Independent of the number of cores, the computational effort

for calculating perturbations is low compared to running a model simulation. Despite small variations, the computing time increases linearly with the number of eigenmodes calculated. The required computational effort appears to increase approxi-





**Table 1.** Computation time for the solution of the eigenproblem in the KL ensemble algorithm for representative setups. The dimension of the system was reduced as described in Sect. 3. The relative computing time is given w.r.t. 8 eigenmodes from 32 combined sensitivities of 5 parameters ("*"). The last two lines show the setups of independent and combined sensitivities presented in this study, respectively.

| # parameters | # sensitivities | # eigenmodes | time | relative time |
|:---:|:---:|:---:|:---:|:---:|
| 5 | 32 (full) | 8 | 660 sec | * |
| **1** | 32 (full) | 8 | 22 sec | 0.03 |
| 5 | **8** (full) | 8 | 122 sec | 0.18 |
| 5 | 8 (**indep.**) | 8 | 148 sec | 0.22 |
| 5 | 32 (full) | **16** | 1108 sec | 1.68 |

"# parameters" = number of different model parameters considered, "# sensitivities" = number of sensitivities used in covariance, "# eigenmodes" = number of eigenvalues and -vectors calculated, time = physical time required for computation of perturbations, relative time = computation time divided by reference computation time

mately linearly with the number of sensitivities for covariance construction and the number of calculated eigenmodes. In this case, doubling the number of considered sensitivities increases the computing time by about a factor of 2.3. Applying the assumption of independent sensitivities reduces the number of used sensitivities from 32 full to 8 independent sensitivities. In addition to the strong reduction of computation time for simulating the sensitivities, the time for solving the eigenproblem reduces by about a factor of 4.5.

The potential of the KL ensemble algorithm is investigated for regional forecasts of a set of biogenic emissions. During the selected case study in the Po valley in July 2012, biogenic emissions were exceptionally sensitive to several land surface properties. In this case, the eigenmode decomposition indicates high correlations of uncertainties in the regional domain as well as between different biogenic gases. Rapidly decreasing eigenvalues state the dominant contributions of only a few orthogonal components from a global point of view. Resulting perturbation factors for isoprene emissions created by the KL ensemble algorithm range between less than 0.1 up to 10. Although some realizations show common perturbation patterns, significant contributions from the subsequent eigenmodes can clearly be identified from the eight realizations. This indicates that the KL ensemble approach is able to sufficiently sample the subspace of leading uncertainties by only as low as 10 members in this case. Moreover, as each eigenmode represents common patterns of different sensitivities, the realizations are affected by the whole set of underlying sensitivities.

The presented application of the KL ensemble generation algorithm demonstrates its potential for an efficient estimation of forecast uncertainties induced by model parameters in high-dimensional atmospheric models. Specifically, the presented algorithm allows for (1) an efficient estimation of various sensitivities based on the assumption of independent sensitivities, (2) an efficient extraction of leading coupled uncertainties using highly parallelized eigenmode decomposition, and (3) an efficient generation of perturbations of high-dimensional parameter fields by the Karhunen-Loéve expansion. This motivates its promising application to various state-dependent parameters in chemistry transport modeling and potentially also in other



atmospheric models. A follow up study will investigate and validate probabilistic forecasts of biogenic gases with respect to different state-dependent model parameters during the PEGASOS campaign in Po valley 2012. Furthermore, the approach

may also be applied to other kinds of model parameters, where sufficient covariances need to be estimated accordingly. In this context, the reduction to leading coupled uncertainties offers the ability to account for dominant uncertainties across all parameters influencing atmospheric chemical forecasts. This would provide a significant step in the transition from deterministic to probabilistic chemistry transport modeling.

*Code availability.*    The code of the KL ensemble algorithm available at https://doi.org/10.5281/zenodo.4468571 (Vogel and Elbern, 2021)

contains the routines which are important for generating the results presented in this study. The routines are embedded in the EURAD-IM system which is stored locally at the Rhenish Institute for Environmental Research as well as at the Jülich Supercomputer Centre (JSC) of Research Centre Jülich.

*Author contributions.*    AV developed and implemented the algorithm, performed the simulations and wrote the manuscript. HE provided the basic idea, supervised the work, contributed to the developments and helped in the preparation of the manuscript.

*Competing interests.*    The authors declare that they have to competing interests.

*Acknowledgements.*    This work has been funded by the Helmholtz Climate Initiative REKLIM (Regional Climate Change), a joint research project of the Helmholtz Association of German research centers (HGF) under grant: REKLIM-2009-07-16. The authors gratefully acknowledges the computing time granted through JARA-HPC on the supercomputer JURECA (Jülich Supercomputing Centre, 2018) at Forschungszentrum Jülich. The authors thank the Institute of Geophysics and Meteorology at the University of Cologne for the opportunity

to finish the work presented here.

**Appendix A: Notations and Variables**





**Table A1.** List of notations used in the formulation of the algorithm including examples of the application to biogenic emissions.

| Term | Expression | Description | Examples |
|---|---|---|---|
| **(Model) Argument** | $i \in [1, I]$ | Arguments in the model configuration including the specification of initial conditions, input fields and model parameterizations. | $I = 6$: land use information / global meteorology / land surface model / boundary layer- / microphysics- / radiation parameterization |
| **Implementation** | $r_i \in [1, R_i]$ | Available options of each model argument $i$. | $R_i = 2 \;\; \forall i$: e.g. $r_1 = 1 \to$ USGS / $r_1 = 2 \to$ MODIS, $r_2 = 1 \to$ ECMWF / $r_2 = 2 \to$ GFS, $r_3 = 1 \to$ Pleim-Xiu / $r_3 = 2 \to$ RUC, $\dots$ |
| **Reference Implementation** | $r_{i*}$ | Selected reference implementation of each model argument. | $r_{i*} = 1 \, \forall i$ (USGS, ECMWF, Pleim-Xiu, ...) |
| **(Model) Setup** | $\{r_1, r_2, \dots, r_I\}$ $j \in [1, J]$ | Specific set of implementations $r_i$ of all model arguments $i$. Index indicating one specific model setup. | — e.g. $\{1,1,1,1,1,1\}$ / $\{1,1,1,1,1,2\}$ / $\{1,1,1,1,2,1\}$ / ... / $\{2,2,2,2,2,2\}$ — e.g. $j = 1 \to \{1,1,1,1,1,1\}$ / $j = 2 \to \{1,1,1,1,1,2\}$ / ... / $j = J \to \{2,2,2,2,2,2\}$ |
| **Reference Setup** | $j_* \to$ $\{r_{1*}, r_{2*}, \dots, r_{I*}\}$ | Index of reference setup representing the set of reference implementations $r_{i*}$ of all model arguments $i$. | $j_* = 1 \to \{1,1,1,1,1,1\}$ |
| **Dimension** | $N$ | Dimension of the problem (total number of elements in the set of considered model parameters). | 5 parameters at 6.572 land surface grid-boxes $\Rightarrow$ $N = 32.860$ |
| **Index Set** | $S = \{s_1, s_2, \dots, s_N\}$ | Set of indices representing the positions of all perturbed model parameters $p$ at all grid boxes $(x, y, z)$. | $S = \Big\{(p_1, x_1, y_1, z_1), (p_2, x_1, y_1, z_1),$ $(p_3, x_1, y_1, z_1), \dots \Big\}$ |
| **Position** | $s \in S$ | Element in the index set representing the position of model parameter $p_n$ at grid box $(x_n, y_n, z_n)$. | e.g. $s_1 = (p_1, x_1, y_1, z_1)$ / $s_2 = (p_2, x_1, y_1, z_1)$ / $s_3 = (p_3, x_1, y_1, z_1)$ / ... |
| **(Model) Parameter** | $Q_j^s(t)$ $q_i^s(r_i, t)$ | Parameter value from model setup $j$ at time $t$ and position $s$. Parameter value from setup were only the $i$-th argument differs from reference setup $r_i \neq r_{i*}$. $\big\{q_i^s(r_i, t)\big\} \subset \big\{Q_j^s(t)\big\}$ | Biogenic emission strength from model setup $j$ at time $t$ and position $s$ |
| **Sensitivity Factor** — **Independent** | $F_j^s$ $f_i^s(r_i)$ | Temporally averaged amplification factor of model setup $j$ at position $s$ w.r.t. the reference setup $j_*$ (see Eq. (1) ). Sensitivity factor w.r.t. single model argument $i$ with implementation $r_i \neq r_{i*}$ differing from the reference (assumed to be independent of other arguments). | |
| **Sensitivity** — **Independent** | $X_j^s$ $x_i^s(r_i)$ | Sensitivity to model setup $j$ at position $s$ (see Eq. (2) ). Sensitivity w.r.t. single model argument $i$ with $r_i \neq r_{i*}$ (assumed to be independent of other arguments). | |
| **Perturbation** | $Y_{\omega_p}^s$ | Perturbation as $p$-th random realization of the KL expansion at position $s$ (see Eq. (15) ). | |
| **Perturbation Factor** | $F_{\omega_p}^s$ | Perturbation factor to be applied to the model parameter at position $s$ of the $p$-th member in the ensemble forecast. | Emission factor of the $p$-th member multiplied to biogenic emissions at position $s$. |



## Appendix B: Derivation of Independent Mean and Covariance

Given a set of independent sensitivities $\left\{x_i^s(r_i)\right\}_{\substack{r_i \in [1,R_i] \\ i \in [1,I]}}$ with implementation $r_i \in [1, R_I]$ of each model argument $i \in [1, I]$ at position $s \in S$. Assuming independence of sensitivities, each combined sensitivity $X_j^s$ is given by

$$X_j^s := X_{\{r_1, r_2, \cdots, r_I\}}^s = \sum_{i=1}^{I} x_i^s(r_i) \qquad \forall\, j \in [1, J]\, , s \in S \quad , \tag{B1}$$

where the total number of combined sensitivities is

$$J := \prod_{i=1}^{I} R_i \quad . \tag{B2}$$

The **mean value** $\mu(s)$ of all combined sensitivities $\left\{X_j^s\right\}_{j \in [1,J]}$ at position $s \in S$ can be calculated from the set of independent sensitivities $\left\{x_i^s(r_i)\right\}_{\substack{r_i \in [1,R_i] \\ i \in [1,I]}}$ as follows:





$$\mu(s) := \frac{1}{J}\sum_{j=1}^{J}X_j^s \overset{(B2)}{=} \frac{1}{\prod_{i=1}^{I}R_i}\sum_{r_1=1}^{R_1}\sum_{r_2=1}^{R_2}\cdots\sum_{r_I=1}^{R_I}X_{\{r_1,r_2,\ldots,r_I\}}^s \overset{(B1)}{=} \frac{1}{\prod_{i=1}^{I}R_i}\sum_{r_1=1}^{R_1}\sum_{r_2=1}^{R_2}\cdots\sum_{r_I=1}^{R_I}\left(\sum_{i=1}^{I}x_i^s(r_i)\right)$$

$$= \frac{1}{J}\Bigg(\Big(x_1^s(1)+x_2^s(1)+\cdots+x_I^s(1)\Big)+\Big(x_1^s(1)+x_2^s(1)+\cdots+x_I^s(2)\Big)+\cdots+\Big(x_1^s(1)+x_2^s(1)+\cdots+x_I^s(R_I)\Big)$$

$$+\ldots \qquad \vdots \qquad\qquad \vdots \qquad\qquad \vdots$$

$$+\Big(x_1^s(1)+x_2^s(2)+\cdots+x_I^s(1)\Big)+\Big(x_1^s(1)+x_2^s(2)+\cdots+x_I^s(2)\Big)+\cdots+\Big(x_1^s(1)+x_2^s(2)+\cdots+x_I^s(R_I)\Big)$$

$$+\ldots \qquad \vdots \qquad\qquad \vdots \qquad\qquad \vdots$$

$$+\Big(x_1^s(1)+x_2^s(R_2)+\cdots+x_I^s(1)\Big)+\Big(x_1^s(1)+x_2^s(R_2)+\cdots+x_I^s(2)\Big)+\cdots+\Big(x_1^s(1)+x_2^s(R_2)+\cdots+x_I^s(R_I)\Big)$$

$$+\Big(x_1^s(2)+x_2^s(1)+\cdots+x_I^s(1)\Big)+\Big(x_1^s(2)+x_2^s(1)+\cdots+x_I^s(2)\Big)+\cdots+\Big(x_1^s(2)+x_2^s(1)+\cdots+x_I^s(R_I)\Big)$$

$$+\ldots \vdots \qquad\qquad \vdots \qquad\qquad \vdots$$

$$+\Big(x_1^s(R_i)+x_2^s(1)+\cdots+x_I^s(1)\Big)+\Big(x_1^s(R_i)+x_2^s(1)+\cdots+x_I^s(2)\Big)+\cdots+\Big(x_1^s(R_1)+x_2^s(1)+\cdots+x_I^s(R_I)\Big)$$

$$+\ldots \qquad \vdots \qquad\qquad \vdots \qquad\qquad \vdots$$

$$+\Big(x_1^s(R_1)+x_2^s(R_2)+\cdots+x_I^s(R_I)\Big)+\Big(x_1^s(R_1)+x_2^s(R_2)+\cdots+x_I^s(R_I)\Big)+\cdots+\Big(x_1^s(R_1)+x_2^s(R_2)+\cdots+x_I^s(R_I)\Big)\Bigg)$$

$$= \frac{1}{\prod_{i=1}^{I}R_i}\left[\left(\prod_{i=2}^{I}R_i\right)\cdot\left(\sum_{r_1=1}^{R_1}x_1^s(r_1)\right)+\left(\prod_{i=1|i\neq 2}^{I}R_i\right)\cdot\left(\sum_{r_2=1}^{R_2}x_2^s(r_2)\right)+\cdots+\left(\prod_{i=1}^{I-1}R_i\right)\cdot\left(\sum_{r_I=1}^{R_I}x_I^s(r_I)\right)\right]$$

$$= \frac{1}{R_1}\cdot\left(\sum_{r_1=1}^{R_1}x_1^s(r_1)\right)+\frac{1}{R_2}\cdot\left(\sum_{r_2=1}^{R_2}x_2^s(r_2)\right)+\cdots+\frac{1}{R_I}\cdot\left(\sum_{r_I=1}^{R_I}x_I^s(r_I)\right)$$

$$= \sum_{i=1}^{I}\left(\frac{1}{R_i}\cdot\sum_{r_i=1}^{R_i}x_i^s(r_i)\right) \tag{B3}$$

The **covariance** $C(s,s')$ of all combined sensitivities $\left\{X_j^{s/s'}\right\}_{j\in[1,J]}$ at positions $s/s' \in S$, respectively, can be calculated

from the set of independent sensitivities $\left\{x_i^{s/s'}(r_i)\right\}_{\substack{r_i\in[1,R_i]\\ i\in[1,I]}}$ as follows:





$$
C(s,s') := \frac{1}{J-1} \sum_{j=1}^{J} \left( X_j^s - \mu^s \right) \left( X_j^{s'} - \mu^{s'} \right) = \frac{1}{J-1} \left[ \sum_{j=1}^{J} \left( X_j^s \cdot X_j^{s'} \right) - \mu^s \left( \sum_{j=1}^{J} X_j^{s'} \right) - \mu^{s'} \left( \sum_{j=1}^{J} X_j^s \right) + J \cdot \mu^s \cdot \mu^{s'} \right]
$$

$$
= \frac{1}{J-1} \left[ \sum_{j=1}^{J} \left( X_j^s \cdot X_j^{s'} \right) - \mu^s \left( J \cdot \mu^{s'} \right) - \mu^{s'} \left( J \cdot \mu^s \right) + J \cdot \mu^s \cdot \mu^{s'} \right] = \frac{1}{J-1} \left[ \sum_{j=1}^{J} \left( X_j^s \cdot X_j^{s'} \right) - J \cdot \mu^s \cdot \mu^{s'} \right]
$$

$$
\overset{(B2)}{=} \frac{1}{J-1} \sum_{r_1=1}^{R_1} \sum_{r_2=1}^{R_2} \cdots \sum_{r_I=1}^{R_I} \left( X_{\{r_1,r_2,\ldots,r_I\}}^s \cdot X_{\{r_1,r_2,\ldots,r_I\}}^{s'} \right) - \frac{J}{J-1} \cdot \mu^s \cdot \mu^{s'}
$$

$$
\overset{(B1)}{=} \frac{1}{J-1} \sum_{r_1=1}^{R_1} \sum_{r_2=1}^{R_2} \cdots \sum_{r_I=1}^{R_I} \left[ \left( \sum_{i=1}^{I} x_i^s(r_i) \right) \cdot \left( \sum_{i=1}^{I} x_i^{s'}(r_i) \right) \right] - \frac{J}{J-1} \cdot \mu^s \cdot \mu^{s'}
$$


$$
\overset{(B3)}{=} \frac{1}{J-1} \sum_{i=1}^{I} \left[ \underbrace{\left( \prod_{l=1|l\neq i}^{I} R_l \right) \cdot \sum_{r_i=1}^{R_i} \left( x_i^s(r_i) \cdot x_i^{s'}(r_i) \right)}_{\text{quadratic terms}} + \underbrace{\sum_{k=1|k\neq i}^{I} \left[ \left( \prod_{l=1|l\neq i,l\neq k}^{I} R_l \right) \cdot \sum_{r_i=1}^{R_i} \sum_{r_k=1}^{R_k} \left( x_i^s(r_i) \cdot x_k^{s'}(r_k) \right) \right]}_{\text{non-quadratic terms}} \right]
$$

$$
- \frac{J}{J-1} \left[ \sum_{i=1}^{I} \left( \frac{1}{R_i} \cdot \sum_{r_i=1}^{R_i} x_i^s(r_i) \right) \right] \cdot \left[ \sum_{i=1}^{I} \left( \frac{1}{R_i} \cdot \sum_{r_i=1}^{R_i} x_i^{s'}(r_i) \right) \right]
$$

$$
\overset{(B2)}{=} \frac{J}{J-1} \sum_{i=1}^{I} \left[ \frac{1}{R_i} \cdot \sum_{r_i=1}^{R_i} \left( x_i^s(r_i) \cdot x_i^{s'}(r_i) \right) + \sum_{k=1|k\neq i}^{I} \left[ \frac{1}{R_i \cdot R_k} \cdot \sum_{r_i=1}^{R_i} \sum_{r_k=1}^{R_k} \left( x_i^s(r_i) \cdot x_k^{s'}(r_k) \right) \right] \right]
$$

$$
- \frac{J}{J-1} \sum_{i=1}^{I} \left[ \frac{1}{(R_i)^2} \cdot \sum_{r_i=1}^{R_i} \left( x_i^s(r_i) \cdot x_i^{s'}(r_i) \right) + \sum_{k=1|k\neq i}^{I} \left[ \frac{1}{R_i \cdot R_k} \cdot \sum_{r_i=1}^{R_i} \sum_{r_k=1}^{R_k} \left( x_i^s(r_i) \cdot x_k^{s'}(r_k) \right) \right] \right]
$$

$$
= \frac{J}{J-1} \sum_{i=1}^{I} \left[ \left( \frac{1}{R_i} - \frac{1}{(R_i)^2} \right) \cdot \sum_{r_i=1}^{R_i} \left( x_i^s(r_i) \cdot x_i^{s'}(r_i) \right) \right] \tag{B4}
$$

**Appendix C: Setup Combined Sensitivities**





**Table C1.** Setups of combined sensitivities. Deviations from the reference setup (denoted as **\***) are give in bold letters ('PX' = Pleim-Xiu surface layer parameterization, 'Du' = Dudhia shortwave radiation parameterization).

| land use | number | global | land surface | boundary layer | microphysics | radiation |
|---|---|---|---|---|---|---|
| *USGS* | 1 * | *ECMWF* | *Pleim-Xiu* | *MYJ+Eta* | *WSM6* | *RRTMG* |
| *USGS* | 2 | *ECMWF* | **RUC** | *MYJ+Eta* | *WSM6* | *RRTMG* |
| *USGS* | 3 | *ECMWF* | *Pleim-Xiu* | **ACM2+PX** | *WSM6* | *RRTMG* |
| *USGS* | 4 | *ECMWF* | *Pleim-Xiu* | *MYJ+Eta* | **TGS** | *RRTMG* |
| *USGS* | 5 | *ECMWF* | *Pleim-Xiu* | *MYJ+Eta* | *WSM6* | **Du+RRTM** |
| *USGS* | 6 | *ECMWF* | **RUC** | **ACM2+PX** | *WSM6* | *RRTMG* |
| *USGS* | 7 | *ECMWF* | *Pleim-Xiu* | *MYJ+Eta* | **TGS** | **Du+RRTM** |
| *USGS* | 8 | *ECMWF* | **RUC** | **ACM2+PX** | **TGS** | **Du+RRTM** |
| *USGS* | 9 | **GFS** | *Pleim-Xiu* | *MYJ+Eta* | *WSM6* | *RRTMG* |
| *USGS* | 10 | **GFS** | **RUC** | *MYJ+Eta* | *WSM6* | *RRTMG* |
| *USGS* | 11 | **GFS** | *Pleim-Xiu* | **ACM2+PX** | *WSM6* | *RRTMG* |
| *USGS* | 12 | **GFS** | *Pleim-Xiu* | *MYJ+Eta* | **TGS** | *RRTMG* |
| *USGS* | 13 | **GFS** | *Pleim-Xiu* | *MYJ+Eta* | *WSM6* | **Du+RRTM** |
| *USGS* | 14 | **GFS** | **RUC** | **ACM2+PX** | *WSM6* | *RRTMG* |
| *USGS* | 15 | **GFS** | *Pleim-Xiu* | *MYJ+Eta* | **TGS** | **Du+RRTM** |
| *USGS* | 16 | **GFS** | **RUC** | **ACM2+PX** | **TGS** | **Du+RRTM** |
| **MODIS** | 1 | *ECMWF* | *Pleim-Xiu* | *MYJ+Eta* | *WSM6* | *RRTMG* |
| **MODIS** | 2 | *ECMWF* | **RUC** | *MYJ+Eta* | *WSM6* | *RRTMG* |
| **MODIS** | 3 | *ECMWF* | *Pleim-Xiu* | **ACM2+PX** | *WSM6* | *RRTMG* |
| **MODIS** | 4 | *ECMWF* | *Pleim-Xiu* | *MYJ+Eta* | **TGS** | *RRTMG* |
| **MODIS** | 5 | *ECMWF* | *Pleim-Xiu* | *MYJ+Eta* | *WSM6* | **Du+RRTM** |
| **MODIS** | 6 | *ECMWF* | **RUC** | **ACM2+PX** | *WSM6* | *RRTMG* |
| **MODIS** | 7 | *ECMWF* | *Pleim-Xiu* | *MYJ+Eta* | **TGS** | **Du+RRTM** |
| **MODIS** | 8 | *ECMWF* | **RUC** | **ACM2+PX** | **TGS** | **Du+RRTM** |
| **MODIS** | 9 | **GFS** | *Pleim-Xiu* | *MYJ+Eta* | *WSM6* | *RRTMG* |
| **MODIS** | 10 | **GFS** | **RUC** | *MYJ+Eta* | *WSM6* | *RRTMG* |
| **MODIS** | 11 | **GFS** | *Pleim-Xiu* | **ACM2+PX** | *WSM6* | *RRTMG* |
| **MODIS** | 12 | **GFS** | *Pleim-Xiu* | *MYJ+Eta* | **TGS** | *RRTMG* |
| **MODIS** | 13 | **GFS** | *Pleim-Xiu* | *MYJ+Eta* | *WSM6* | **Du+RRTM** |
| **MODIS** | 14 | **GFS** | **RUC** | **ACM2+PX** | *WSM6* | *RRTMG* |
| **MODIS** | 15 | **GFS** | *Pleim-Xiu* | *MYJ+Eta* | **TGS** | **Du+RRTM** |
| **MODIS** | 16 | **GFS** | **RUC** | **ACM2+PX** | **TGS** | **Du+RRTM** |

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
