# Peer review of "Efficient ensemble generation for uncertain correlated parameters in atmospheric chemical models: a case study for biogenic emissions from EURAD-IM version 5"

_Geoscientific Model Development, 2021_

## Referee Comment (RC1)

Review of "**Efficient ensemble generation for uncertain correlated parameters in atmospheric chemical models**"

The authors presents a systematic and efficient approach to generate emission perturbations in chemical transport models. They applied their method to biogenic emissions of isoprene (in particular). As they pointed out, perturbing emissions can easily require an extremely large number of perturbations. First they derive an important result that under linearity assumption (or TLM) the sensitivity of perturbed model parameters when considering all combinations scales as $R^{J}$ where J is the number of model parameters and R the number of cases considered (generally 2 – one for control the other for the perturbed. It is shown that under the assumption of linearity, that this total sensitivity can be obtained by perturbing each model components individually, this significantly reducing the total number of perturbation. Secondly by constructing the error covariance of those perturbations (in model space), it is possible to obtain the leading eigenvalues/eigenmodes of the covariance matrix by using a large scale eigenvalue software known as APRACK that does not require storing a (very large) covariance matrix. Finally, an approximation of the perturbations can be obtained from these leading eigenvalues/eigenvectors by using a Karhunen-Loeve expansion. Although, as the authors discusses, the equivalence between the combined sensitivities and the independent sensitivities are not meet with certain nonlinear processes, such as meteorology, this is nevertheless an important contribution that should be published. However, the document somewhat hard to follow, especially in the mathematical description of the method, and some clarifications and a rearrangement of the theory would be most beneficial. Although this may represent some rewriting, it is believed that it can be easily accomplished.

*Major issue*:

Section 2.1 is hard to follow and uses concepts that are not well defined. Table A1 gives examples that greatly help understand what the concepts may actually mean. It is unfortunate that this table appears in the appendix section. The authors should provide examples (as in Table 1) of the concepts introduced – especially for the second paragraph Lines 105 to 111. It maybe worth considering splitting section 2 into a section on "Efficient sensitivity calculation" using sections 2.1, 2.1.1, and then 2.1.2, followed by a new section (section 3) discussing the algorithm which would include lines 82-91, figure 1, section 2.2 and 2.3. It could also be welcomed to have a figure in the section 2 (around equation 9) that shows the required number of forecasts J as a function of I for the combined sensitivity and independent sensitivity calculation (for a few values of R).

*Minor points*:

1 – line 67-69. Should draw the parallel and differences between the Principal component analysis and the Karhunen-Loeve expansion for discrete functions. Also the Principal component analysis is in fact widely used in geophysical sciences for example in climatology.

2 – line 98. It is not clear what is multi-variational covariance ? Do you mean multivariate covariance ? If not this has to be defined.

3 – line 203.  Please define s/s'

4 – line 262.  Is the joint perturbation is define by using a multivariate covariance matrix, C.  Or is it an observation the results for each species leads to similar eigenvectors ?

5 – Lines 333-337.  The results not shown, should be shown as it is part of the main finding of this approach and study.

6 – Figure 5.  Not sure what the numbers above each panel refers to.  Please explain or drop.  Same for figure 7.

7- Lines 355-356.  Don't quite follow the argument of this sentence "The approach is based  … " .  Please expand and explain.

8 – Section 4. Discussion and conclusions.  Could you comment on how this method may provide an error uncertainty associated with each eigenvector of the expansion, and thus how the method could be used in inverse modelling.

---

## Author Comment (AC1)

**Response to Chief Editor (Astrid Kerkweg):**

Dear authors,
in my role as Executive editor of GMD, I would like to bring to your attention our Editorial version 1.2:
https://www.geosci-model-dev.net/12/2215/2019/
This highlights some requirements of papers published in GMD, which is also available on the GMD website in the 'Manuscript Types' section:
http://www.geoscientific-model-development.net/submission/manuscript_types.html

General Reply: We are grateful for the detailed information and suggestions and hope that the manuscript does now fulfill all criteria.

In particular, please note that for your paper, the following requirements have not been met in the Discussions paper:

- "The main paper must give the model name and version number (or other unique identifier) in the title."

- "If the model development relates to a single model then the model name and the version number must be included in the title of the paper. If the main intention of an article is to make a general (i.e. model independent) statement about the usefulness of new development, but the usefulness is shown with the help of one specific model, the model name and version number must be stated in the title. The title could have a form such as, "Title outlining amazing generic advance: a case study with Model XXX (version Y)"."

As you are using EURAD-IM to show the performance of the ensemble, please add something like "a case study using EURAD-IM version x.y" to the title of your manuscript.

Reply1: We thank the chief editor for pointing towards the specific requirement. Indeed, this article presents a general statement about the usefulness of new development, but the usefulness is shown with the help of one specific model. We added the suggested information to the title:
*" a case study for biogenic emissions from EURAD-IM version 5 "*

Additionally, please note, that as you are using EURAD-IM to produce the results shown in your article, the information how to access the EURAD-IM code is also required, including the permanent archiving of the exact EURAD-IM version the results of this articles have been created with.

Reply2: To be specific, the EURAD-IM model itself is not required. The developments presented in this article does only require the output of an chemical transport model, which is the EURAD-IM model for the presented results. We see that the formulation of this aspect was misleading in the article and reformulated the related sentences accordingly (Sect. 3, l.305-307 and l.314-315, new count):
*" The KL ensemble generation algorithm was implemented in a way that it uses precalculated output from the EURAD-IM (EURopean Air pollution Dispersion - Inverse Model) chemical data assimilation system. Note that the algorithm is independent of the forecast model, which can be replaced by any other CTM. [. . .] In this study, the EURAD-IM system provides forecasts of sensitivities to various model arguments, which are used for covariance construction in the KL algorithm. "*

Additionally, we made the output of EURAD-IM which was used for the calculation of the results available. The code availability statement was extended accordingly to a code and data availability

statement including:

*" The data used as input for the production of the results and the output of the algorithm are available at https://doi.org/10.5281/zenodo.4772909 (Vogel and Elbern, 2021c). "*

---

## Author Response (AR1)

Response to Reviewer 1:
We are grateful for the insightful remarks which we all happy to accept and hope that we could address them all in a satisfying way.

The authors presents a systematic and efficient approach to generate emission perturbations in chemical transport models. They applied their method to biogenic emissions of isoprene (in particular). As they pointed out, perturbing emissions can easily require an extremely large number of perturbations. First they derive an important result that under linearity assumption (or TLM) the sensitivity of perturbed model parameters when considering all combinations scales as $R^J$ where **J** is the number of model parameters and **R** the number of cases considered (generally **2** – one for control the other for the perturbed. It is shown that under the assumption of linearity, that this total sensitivity can be obtained by perturbing each model components individually, this significantly reducing the total number of perturbation. Secondly by constructing the error covariance of those perturbations (in model space), it is possible to obtain the leading eigenvalues/eigenmodes of the covariance matrix by using a large scale eigenvalue software known as APRACK that does not require storing a (very large) covariance matrix. Finally, an approximation of the perturbations can be obtained from these leading eigenvalues/eigenvectors by using a Karhunen-Loeve expansion. Although, as the authors discusses, the equivalence between the combined sensitivities and the independent sensitivities are not meet with certain nonlinear processes, such as meteorology, this is nevertheless an important contribution that should be published. However, the document somewhat hard to follow, especially in the mathematical description of the method, and some clarifications and a rearrangement of the theory would be most beneficial. Although this may represent some rewriting, it is believed that it can be easily accomplished.

Major issue:
Section 2.1 is hard to follow and uses concepts that are not well defined. Table A1 gives examples that greatly help understand what the concepts may actually mean. It is unfortunate that this table appears in the appendix section. The authors should provide examples (as in Table 1) of the concepts introduced – especially for the second paragraph Lines 105 to 111. It maybe worth considering splitting section 2 into a section on "Efficient sensitivity calculation" using sections 2.1, 2.1.1, and then 2.1.2, followed by a new section (section 3) discussing the algorithm which would include lines 82-91, figure 1, section 2.2 and 2.3.

Reply: We thank the reviewer for pointing out this difficulty. As this aspect is in accordance to Reviewer 2, we are happy to thoroughly revise the description of the algorithm. As suggested, we divided Sect 2 into two separate sections were we first provide an extended description of the mathematical basis of the sensitivity calculations (new Sect.2: "Concept of Sensitivity Estimation") including Tab. A1 (now Tab. 1) and examples. This section is followed by a compressed formulation of the algorithm (new Sect. 3). Most of the descriptive text and formulas were not affected by the rearrangements. The introduction of the new Sect. 2 was formulated as follows (l.92-96, new count):

*" This section introduces the conceptual basis for the description of the ensemble generation algorithm in Sect. 3. The algorithm relies on several definitions which are introduced in Sect. 2.1. Given these definitions, the concept of sensitivities consists of two parts: the general formulation of sensitivities in Sect. 2.2, and the special formulation of independent sensitivities in Sect. 2.3. Each of these parts provides the basis for combined or independent covariance construction in the ensemble generation algorithm, respectively. "*

Some minor modifications of the existing parts of the manuscript were made, which did not modify the scientific content:

- Note about the new Sect. 2 in the introduction (l.86, new count): *" Section 2 provides the concept of sensitivity estimation on which the ensemble generation approach is based. "*

- Update of the name and of the first step in the algorithm-section. As the sensitivity estimation is now covered by the new Sect. 2, the first step of the algorithm is now named "Covariance Construction" to avoid confusion (l.193, l.199, Fig. 2, new count).

- References to the related parts of the new Sect. 2 and the overview over important terms in Tab. 1 in the condensed Sect. 3.

- Include references in the description of the results (new Sect. 4) w.r.t related parts of the new Sect. 2 and Sect. 3.

Additionally, the rearrangement was also used to make some points clearer as suggested by both reviewers. The new parts with major modifications include:

- Extension of the description of used terms, which was put into a separate subsection (new Sect. 2.1) and consists of a much more detailed description (as requested by Reviewer 2) including specific examples (as requested by Reviewer 1). The Subsection reads now (l.98-120, new count):

  *" The concept of sensitivity estimation requires the definition of several terms. This section introduces these terms on a general level and provides examples for the application to CTMs. All important terms used in the concept of sensitivity estimation and the algorithm are summarized in Tab. 1, including specific examples for the application to biogenic emissions.*

  *Generally, the term model parameter refers to any parameter in the prognostic equations of the model which may affect the models forecast. A prominent example of highly uncertain model parameters in CTMs are trace gas emissions. Considering multiple model parameters, like the emission rates of different trace gases, the dimension $N$ of the problem is the total number of all considered parameters at all grid-boxes. The total set of all parameter values at all grid-boxes at time $t$ is denoted by vector $Q(t) \in \mathcal{R}^N$. In the case of trace gas emissions, $Q(t)$ includes the simulated emission rates of all considered gases at all grid-boxes. Thus, the n-th entry $Q^{s_n}(t)$ of the parameter vector is the simulated value of model parameter $p_n$ at grid-box $(x_n, y_n, z_n)$. The index $s_n = (p_n, x_n, y_n, z_n)$ specifies the model parameter and grid-box and is therefore denoted as position. Hence, the positions of all parameters at all grid-boxes is given by the index set $S := \{s_1, s_2, \ldots, s_N\}$.*

  *The concept of sensitivities uses a set of $J$ parameter vectors $Q_j(t)$ from differently configured model simulations $j \in [1, J]$. In this approach, different model simulations are achieved by using different implementations $r_i$ of a set of model arguments $i \in [1, I]$. The term model argument comprises a heterogeneous set of available arguments for the specific configuration of the model. In this regard, model arguments in CTMs may be as diverse as initial and boundary conditions, any external input fields and the formulation of parameterizations in the model. The specific implementation $r_i$ of a model argument $i$ is realized by selecting one available option of the argument in the model. For example, input fields of land surface properties may be one model argument with two implementations: land surface information from source A and from source B. In the concept of sensitivity estimation, each model argument $i \in [1, I]$ is interpreted as arbitrary parameter with $R_i$ different implementations $r_i \in [1, R_i]$. Thus, each setup index $j \in [1, J]$ represents a complete model setup $\{r_1, r_2, \ldots, r_I\}$ as specific combination of implementations $r_i$ of each model argument $i \in [1, I]$. Then $Q_j^s(t)$ is the parameter value of position $s \in S$ simulated with model setup $j \in [1, J]$ at discrete time $t \in [t_1, t_T]$. Note that the complete set of $\left\{Q_j^s(t)\right\}_{j \in [1,J]}$ considers all possible combinations of implementations of all model arguments. "*

- The overview table (former Tab. A1, now Tab. 1) has been updated accordingly and is now part of the definition subsection (Sect. 2.1, new count).

- Compressed the formulation of covariances and mean from combined sensitivities (old Eq. 3 and 4) as suggested by Reviewer 2. The highly similar equations 3 and 4 are now compressed to one equation (Eq. 9).

- In accordance to the rearrangements and extended description in new Sect. 2.1, the following sentence was included to explain the scope of sensitivities (new Sect. 2.2, l.128-129, new count):
  *" Since the definition of sensitivities refers to all possible combinations of implementations of all model arguments, the set of $\left\{X_j^s\right\}_{j\in[1,J]}$ is also denoted as set of combined sensitivities. "*

- A more detailed comment on the limitations of the independent assumption for atmospheric model parameters (l.489-497, new count, as suggested by Reviewer 2):
  *" The results presented in this study demonstrate a considerable reduction of required computational resources under the assumption of independent sensitivities. As this method assumes linearity of parameter sensitivities, it may not be a sufficient approximation for all atmospheric parameters. However, the linear assumption relates to sensitivities of the perturbed parameters to model configurations. In other words, nonlinear effects resulting from the combination of different model arguments are disregarded. Note that this does not affect the impact of the parameters on prognostic fields like trace gas distributions and their propagation in time, which is beyond the ensemble generation and may still be highly nonlinear. Thus, the presented approach may also be suitable to be applied to parameters in NWP with highly nonlinear model dynamics. The sufficiency of this approximation for a model parameter in relation to the reduction of computational efforts needs to be evaluated for each specific application setup. "*

- Visualization of number of sensitivities required for combined and independent sensitivities (see reply to next aspect, below).

**It could also be welcomed to have a figure in the section 2 (around equation 9) that shows the required number of forecasts J as a function of I for the combined sensitivity and independent sensitivity calculation (for a few values of R).**

Reply: We are very thankful for this suggestion. We added a visualization (new Fig. 1) of the number of combined and independent sensitivities which, we believe, strikingly demonstrates the benefit of the developments. The following explanations were added to the formulation of combined (new Sect. 2.2) and independent sensitivities (new Sect. 2.3), respectively. Combined Sensitivities (l.139-141, new count):

*" Figure 1 shows the exponential increase of the number of combined sensitivities as function of the number of implementations and arguments from Eq. (3). For example, considering six model arguments ($I = 6$) with two implementations each ($R_i = 2, \forall\, i = [1, I]$), requires $J_{combi} = J = 2^6 = 64$ model executions prior to the ensemble generation. "*

Independent Sensitivities (l.169-174, new count):

*" [...]with $J_{indep} \ll J_{combi} = J$ as shown in Fig. 1. [...] For example, considering $I = 3$ model arguments with $R_i = 5$ implementations each, the total number of $J_{combi} = 5^3 = 125$ combined sensitivities reduces to $J_{indep} = 1 + 3 \cdot 4 = 13$ independent sensitivities. For $I = 10$ model arguments with $R_i = 2$ implementations each, the number of required sensitivities can even be reduced by two orders of magnitude ($J_{indep} = 11$, $J_{combi} = 1024$). "*

**Minor points:**

1. **line 67-69. Should draw the parallel and differences between the Principal component analysis and the Karhunen-Loeve expansion for discrete functions. Also**

[Figure]

Figure 1: Number of required sensitivities as function of the number of arguments $I$. Shown are the required numbers of combined ($J_{combi}$, Eq. (3), green) and independent ($J_{indep}$, Eq. (8), orange) sensitivities for different numbers of implementations $R_i = 2, 3, 4, 5 \quad \forall i \in [1, I]$.

**the Principal component analysis is in fact widely used in geophysical sciences for example in climatology.**

Reply1: This aspect has been added in the introduction, including references to PCA in climatology as requested (l.77-79, new count):

*" The eigenmode analysis required for the KL expansion is equivalent to a principal component analysis (PCA) by singular vector decomposition (SVD). The discrete PCA has been used as diagnosis tool in atmospheric sciences and is most established in climatology (e.g., Hannachi et al., 2007; Galin, 2007; Liu et al., 2014; Guilloteau et al., 2021). "*

2. **line 98. It is not clear what is multi-variational covariance ? Do you mean multi-variate covariance ? If not this has to be defined.**

Reply2: Indeed, "multivariate" is the correct word. We replaced it at all locations were it did a pear.

3. **line 203. Please define s/s'**

Reply3: This is a poor notation from our side. By $\varphi_d(s/s')$ we meant $\varphi_d(s)$ and $\varphi_d(s')$, respectively.

- The notation was reformulated accordingly at l.240-241 (new count):
  *" [...] $\varphi_d(\tilde{s})$ the $\tilde{s}$-th element of the corresponding eigenvector $\varphi_d \in \mathcal{R}^N$ for all $d \in [1, N]$ with $\tilde{s} = s, s' \in S$. "*

- The same notation was used in Eq. (3b) and Eq. (4b) (old count) which now read (new Eq. (9b)):

$$\mu_{\text{combi}}(\tilde{s}) := \frac{1}{J_{\text{combi}}} \sum_{j=1}^{J_{\text{combi}}} X_j^{\tilde{s}} \quad \Big| \quad \tilde{s} = s, s'$$

- A corresponding notation in the Appendix was also adopted the same way (l.594, new count):
  *" The **covariance** $C(s, s')$ of all combined sensitivities $\left\{ X_j^{\tilde{s}} \right\}_{j \in [1, J]}$ at positions $\tilde{s} = s, s' \in S$ can be calculated from the sets of independent sensitivities $\left\{ x_i^{\tilde{s}}(r_i) \right\}_{\substack{r_i \in [1, R_i] \\ i \in [1, I]}}$ as follows: [...] "*

4. **line 262. Is the joint perturbation is define by using a multivariate covariance matrix, C. Or is it an observation the results for each species leads to similar eigenvectors ?**

Reply4: The reviewer is right that the joint perturbation induced by multivariate covariances in C. We added this information in the manuscript (l.325, new count):

*" In the KL algorithm, sensitivities from the five emission fields induce multivariate covariances in the covariance matrix C which allow for joint perturbation of those. "*

5. **Lines 333-337. The results not shown, should be shown as it is part of the main finding of this approach and study.**

Reply5: That is a good point. We extend Fig. 6 (new count) to show the full eigenvectors for combined sensitivities which proofs this statement. The description of the results was adopted accordingly.

6. **Figure 5. Not sure what the numbers above each panel refers to. Please explain or drop. Same for figure 7.**

Reply6: The numbers are the eigenvalues of the corresponding eigenvectors which are displayed as column. This has been clarified in the Figure captions of Fig. 6 (new count):

*" [...] Numbers above each column show the eigenvalues of the corresponding eigenvector. "*

7. **Lines 355-356. Don't quite follow the argument of this sentence "The approach is based ... " . Please expand and explain.**

Reply7: This formulation is maybe misleading. It was reformulated similar to the formulation used in the introduction of Sect. 2, and now reads (l.510, new count):

*" The approach is based on the fact that the forecast model acts as a dynamical system with its spatial and multivariate couplings of the atmospheric state. "*

8. **Section 4. Discussion and conclusions. Could you comment on how this method may provide an error uncertainty associated with each eigenvector of the expansion, and thus how the method could be used in inverse modeling.**

Reply8: We thank the reviewer for this comment and are happy to include a discussion on this in the manuscript (l.498-507, new count):

*" The developed KL approach may extended to inverse optimization of model parameters. The generated parameter ensemble can be used to estimate state-dependent model covariances in an ensemble data assimilation system. If requested by the type of data assimilation algorithm, inverse square roots of covariance matrices are readily available for preconditioned minimization. Furthermore, the KL expansion of the parameter fields enables an advanced optimization approach. Instead of optimizing the parameter fields in its full $N$-dimensional space, the optimization can be performed in the reduced subspace spanned by the $D$ leading eigenmodes. As the leading eigenmodes represent the dominant uncertainties of the parameters, the optimization would be restricted to those. In this case, the set of stochastic coefficients $\{y_d\}_{d \in [1,D]}$ would be replaced by the optimization variable (compare Eq. (15) ) which is fully determined by as low as $D$ observations. Thus, this approach may be able to provide a rough, yet efficient optimization of model parameter fields with a low number of observations. For both optimization approaches, spurious correlations resulting from the restriction to leading uncertainties must be addressed by location measures. "*

**Response to Reviewer 2:**

We thank the reviewer for the thoughtful and detailed evaluation and valuable remarks. We hope that we could reply and adopt the manuscript in a sufficient way.

**Overview:**

The paper presents a method to construct ensembles with a small number (10) of members that still represent the uncertainty in ensemble applications of CTMs. The method consists of the following three steps: a) sensitivity estimation, b) eigenmode decomposition of the sensitivity information and c) generation of the ensemble based on the Karhunen-Loéve expansion. The method is illustrated with the construction of an isoprene emission ensemble derived for one day.

**General remarks:**

The paper contains a derivation of the method in a generalized way and a discussion of its application to create an ensemble of biogenic isoprene emissions. But, it is a considerable weakness of the paper that not enough evidence is given that the ensembles generated by the method would indeed capture the main component of the uncertainty. Without this evidence, the paper has little relevance. Giving such evidence is not trivial. I can only suggest two aspects but there might be other options: 1) use the derived ensemble of isoprene emissions in an actual CTM ensemble application. 2) provide more evidence that the ensembles created using either the combined or independent sensitivities approach lead to similar results.

Reply1: As the authors perceived the concerns of the reviewer differently, we provide a split answer. Because we see two possibilities, we will reply to both of them and hope that this captures the reviewers intention.

a) In case the reviewer asks for an evidence that the ensemble generation method captures the total uncertainties of biogenic trace gas forecasts (including all other efficacious processes affecting the atmospheric distributions of trace gases), we answer in the following way:
This paper does not aim to capture the components of the overall uncertainties of atmospheric chemical forecasts by perturbing biogenic emissions. Certainly it is clear that biogenic emissions engender aerosol formation and contributes to ozone production, the share of which is depending also on the relative contributions from anthropogenic emissions. As a consequence the emission uncertainties are then contributing to the ozone and aerosol uncertainties. This paper presents a demonstrator of a dimension reduction approach for ensembles with the restricted problems of biogenic emission uncertainties. We are certainly aware of the fact that there are many other processes (like atmospheric transport, chemical conversions and deposition to name a few) contribute significantly to forecast uncertainties of biogenic trace gas distributions. A general assessment of sensitivities to uncertainties in these processes is given in the paper Vogel and Elbern, 2021 (https://doi.org/10.5194/acp-21-4039-2021).

b) If the need for an evidence is meant in a specific way - i.e. that the ensemble generation method captures the main uncertainties of the biogenic emission parameters - we totally agree with the reviewers remark. We do see the need for an evaluation and are thankful for the reviewers suggestions in this regard. Ideally, a comprehensive a-posteriori evaluation would be based on a representative amount of data covering multiple conditions. However, this would require longterm simulations in different regions as well as detailed observational data. This induces two problems. Firstly, sufficient observations of biogenic trace gases are rare and limited to special observational campaigns which does not provide a representative amount of data for comprehensive evaluation. Secondly, observations do only provide information on local concentrations - and not emissions. As concentrations are affected by other uncertain

processes, a direct evaluation of an ensemble of emissions by observations would be misleading. Furthermore, a comprehensive evaluation would be a complete study for itself and would expand this publication too much. A note on this was included in the introduction of the results (now Sect. 4, see reply5 below) and in the conclusions (now Sect. 6, l.543-548, new count):

*" A comprehensive evaluation of KL ensemble perturbations would be based on a representative amount of observational data. Comparing to observed trace gas concentrations requires the consideration of uncertainties related to different processes affecting those concentrations which is out of the scope of this study. Instead, the performance of the KL algorithm itself was evaluated using ensemble statistics. The statistical comparison of the KL perturbations with the sensitivities used as input states sufficient representation of the main aspects. Both, combined and independent methods, were able to capture the main uncertainties while smaller contributions were neglected according to the objective of the algorithm. "*

For the evaluation, we decided to follow suggestion 2 of the reviewer and evaluate the final ensemble of perturbations using the assumption of independent sensitivities with the statistics of the uncertainties sampled by the large set of combined sensitivities. As by design, the KL expansion does not change the statistics of the covariance matrix, the statistics of the set of combined sensitivities used for covariance construction present a sufficient basis for the evaluation. We therefore added an evaluation based on a scatter-plot of the ensemble statistics (new Sect. 4.6, l.415-451 and Fig. 10, new count):

*" The performance of the KL ensemble perturbations is evaluated by ensemble statistics. Note that this evaluation does only relate to the algorithm itself - i.e. how well the algorithm is able to capture the uncertainties indicated by the sensitivities. The question on how well the sensitivities represent the true parameter uncertainties is not part of the evaluation. In order to be able to compare the statistics of the KL ensembles using combined and independent sensitivities, $I = 6$ model arguments are considered for both setups. While the setup of 32 combined sensitivities is the same as in the previous sections ($J_{combi} = 32$), no additional unceratinties were included in the independent setup ($J_{indep} = 7$). Despite this, the setup remains as described in Sect. 4.2.*

*In Fig. 10, statistics of the ensemble perturbation factors from the KL algorithm with combined sensitivities are compared to statistics of the sensitivity factors from 32 combined sensitivities. Because these 32 combined sensitivities serve as input for the algorithm in the combined setup, the ensemble statistics at all locations should ideally fall on the identity line. Thus, deviations from the identity line give an indication of the sampling error induced by the limited number of used eigenmodes and the low number of 8 realizations compared to the dimension of the problem. Mean isoprene emission factors for the combined setup are well represented by the KL algorithm, deviations remain below 20% for almost all locations (Fig. 10a).*

*Ensemble standard deviations show more significant deviations (Fig. 10b). For low and medium values, standard deviations of the KL ensemble with combined sensitivities range from 75% until 130% of the respective input values. This almost homogeneous scatter around the input values are likely induced by minor uncertainties which are not captured by the leading eigenmodes. At some locations, the KL ensemble produces a set of high standard deviations between 5.0 and 8.0 which are not found in the sensitivities. While the standard deviation is overestimated at these locations, the ensemble mean at these locations is slightly underestimated (sensitivities approx. 3.0, combined ensemble perturbations approx. 2.0, Fig. 10a). These deviations can be related to the low ensemble size and are expected to reduce for larger ensemble sizes. Additionally, limiting sensitivity factors in the KL algorithm may also affect sensitivity statistics in this case (compare Sect 4.2). The independent setup induces larger deviations of mean isoprene emission factors from the mean combined sensitivities. The increase in deviations mainly represent the additional inaccuracy due to the assumption of independent sensitivities. Note that the 32 combined sensitivities used in the evaluation are a subset of 64 possible combinations (compare Sect. 4.2). Because the independent setup approximates all combinations of sensitivities,*

[Figure]

Figure 2: Scatter plot of ensemble statistics for isoprene emission perturbations. Shown are logarithmic ensemble mean (a) and standard deviations (b) of the KL ensemble perturbations at each grid-box as function of combined sensitivities. Ensemble statistics are shown for 32 combined (green) and 7 independent (orange) sensitivities, which both refer to the same set of $I = 6$ considered arguments. The solid gray line indicates the identity line (e.g. $\mu_{\mathrm{KL}} = 1 \cdot \mu_{\mathrm{sens}}$ for the ensemble mean) to the set of 32 combined sensitivities, the dashed gray lines represent an over- or underestimation by factor 1.5 (e.g. $\mu_{\mathrm{KL}} = 1.5 \cdot \mu_{\mathrm{sens}}$ and $\mu_{\mathrm{KL}} = (1.5)^{-1} \cdot \mu_{\mathrm{sens}}$).

*some deviations might also relate to the selection of the 32 combined sensitivities. Ensemble mean factors correlate well with the combined sensitivities and deviations remain between −25% and +50% for most locations (Fig. 10a). While only 7 instead of 32 â or ideally 64 â sensitivities are required for this setup, the spread of mean mean factors is about twice the spread for the combined ensemble setup. Deviations of ensemble standard deviations are also slightly increased for the independent setup (Fig. 10b). The overestimation of high standard deviations produced by the combined setup is reduced in the independent setup. At some locations, standard deviations of about 2.0 in the combined sensitivities are underestimated by the KL ensemble with independent sensitives (standard deviations approx. 1.5). These differences are likely due to nonlinear effects from combining sensitivities which are neglected in this setup. Nevertheless, the ensemble standard deviations at most locations are well represented by the KL ensemble with independent sensitivities. "*

Of the three steps a), b) and c), a) seems to be the most interesting because it addresses the interesting questions how many model runs are required to capture the sensitivity of model result to variation of model configuration choices. By assuming linearity, the authors derive that only a reference model run and a model run with the a modification of one aspect at a time (i.e. alternative land use map, alternative deposition scheme etc.) is required and that cross-combinations of model configurations (i.e. alternative land use map AND alternative deposition scheme) are not required. Hence, the number of the required runs does not have to be all possible permutations of configuration options but only the number of tested configurations itself. Intuitively speaking, that seems obvious because the sensitivities are assumed to be linear. This means that strong non-linearities as typical for NWP dynamics or atmospheric chemistry may not be suited for the method. A better discussion of the limitations of the choice of linearity is required.

Reply2: We quite agree with the reviewer's claim that the resulting number of configurations is not surprising, and we see the need for some more clarification and discussion in this respect. The linear assumption relates to the effects of model configuration on the perturbed model parameters. Model processes transferring the parameters into other prognostic fields like trace gas distributions and their propagation in time may still be highly nonlinear. Thus, the presented approach may also be suitable to be applied to suitable parameters in NWP. An example could be surface parameters affecting boundary layer dynamics. If such a parameter does depend approximately linearly on properties like vegetation height, uncertainties of this parameter could be sufficiently appropriated by the KL ensemble approach. The atmospheric reaction in terms of thermal or dynamical forcing may still be highly nonlinear following the model dynamics. Sure, further investigation is required to investigate potential candidates of such parameters in NWP. This example aims only to support the argumentation. A discussion of this and related limitations is added to the new discussion Sect. 5 (l.489-497, new count):

*" The results presented in this study demonstrate a considerable reduction of required computational resources under the assumption of independent sensitivities. As this method assumes linearity of parameter sensitivities, it may not be a sufficient approximation for all atmospheric parameters. However, the linear assumption relates to sensitivities of the perturbed parameters to model configurations. In other words, nonlinear effects resulting from the combination of different model arguments are disregarded. Note that this does not affect the impact of the parameters on prognostic fields like trace gas distributions and their propagation in time, which is beyond the ensemble generation and may still be highly nonlinear. Thus, the presented approach may also be suitable to be applied to parameters in NWP with highly nonlinear model dynamics. The sufficiency of this approximation for a model parameter in relation to the reduction of computational efforts needs to be evaluated for each specific application setup. "*

**Section 2, which describes the method (a, b, c) uses a sophisticated mathematical nomenclature, which may be more confusing than enlightening for the typical GMD reader. The chapter is difficult to understand, and it should be made clearer what the main novel ideas are, and what are just mathematical definitions. For example, formulas 3a and b and 4a and b are in my opinions simple repetitions of the well known formulas for covariance and mean. I acknowledge that it is a matter of taste how "mathematical" a paper should be formulated but the mathematical formulae need to convey a specific message relevant to the objective of the paper.**

Reply3: As this aspect is in accordance to Reviewer 1, we see the need for adopting the description of the algorithm. We complied with the suggestion to divide Sect. 2 (old count) into two separate sections were we first provide the mathematical basis of sensitivity estimation (new Sect. 2), followed by a compressed formulation of the algorithm (new Sect. 3). The modification described below also include a reduction of formulas covariances and mean from combined sensitivities (old Eq. 3 and 4).

Some minor modifications of the existing parts of the manuscript were made, which did not modify the scientific content:

- Note about the new Sect. 2 in the introduction (l.86, new count): *" Section 2 provides the concept of sensitivity estimation on which the ensemble generation approach is based. "*

- Update of the name and of the first step in the algorithm-section. As the sensitivity estimation is now covered by the new Sect. 2, the first step of the algorithm is now named "Covariance Construction" to avoid confusion (l.193, l.199, Fig. 2, new count).

- References to the related parts of the new Sect. 2 and the overview over important terms in Tab. 1 in the condensed Sect. 3.

- Include references in the description of the results (new Sect. 4) w.r.t related parts of the new Sect. 2 and Sect. 3.

Additionally, the rearrangement was also used to make some points clearer as suggested by both reviewers. The new parts with major modifications include:

- Extension of the description of used terms, which was put into a separate subsection (new Sect. 2.1) and consists of a much more detailed description (as requested by Reviewer 2) including specific examples (as requested by Reviewer 1). The Subsection reads now (l.98-120, new count):

  " *The concept of sensitivity estimation requires the definition of several terms. This section introduces these terms on a general level and provides examples for the application to CTMs. All important terms used in the concept of sensitivity estimation and the algorithm are summarized in Tab. 1, including specific examples for the application to biogenic emissions.*

  *Generally, the term model parameter refers to any parameter in the prognostic equations of the model which may affect the models forecast. A prominent example of highly uncertain model parameters in CTMs are trace gas emissions. Considering multiple model parameters, like the emission rates of different trace gases, the dimension $N$ of the problem is the total number of all considered parameters at all grid-boxes. The total set of all parameter values at all grid-boxes at time $t$ is denoted by vector $Q(t) \in \mathcal{R}^N$. In the case of trace gas emissions, $Q(t)$ includes the simulated emission rates of all considered gases at all grid-boxes. Thus, the n-th entry $Q^{s_n}(t)$ of the parameter vector is the simulated value of model parameter $p_n$ at grid-box $(x_n, y_n, z_n)$. The index $s_n = (p_n, x_n, y_n, z_n)$ specifies the model parameter and grid-box and is therefore denoted as position. Hence, the positions of all parameters at all grid-boxes is given by the index set $S := \{s_1, s_2, \ldots, s_N\}$.*

  *The concept of sensitivities uses a set of $J$ parameter vectors $Q_j(t)$ from differently configured model simulations $j \in [1, J]$. In this approach, different model simulations are achieved by using different implementations $r_i$ of a set of model arguments $i \in [1, I]$. The term model argument comprises a heterogeneous set of available arguments for the specific configuration of the model. In this regard, model arguments in CTMs may be as diverse as initial and boundary conditions, any external input fields and the formulation of parameterizations in the model. The specific implementation $r_i$ of a model argument $i$ is realized by selecting one available option of the argument in the model. For example, input fields of land surface properties may be one model argument with two implementations: land surface information from source A and from source B. In the concept of sensitivity estimation, each model argument $i \in [1, I]$ is interpreted as arbitrary parameter with $R_i$ different implementations $r_i \in [1, R_i]$. Thus, each setup index $j \in [1, J]$ represents a complete model setup $\{r_1, r_2, \ldots, r_I\}$ as specific combination of implementations $r_i$ of each model argument $i \in [1, I]$. Then $Q_j^s(t)$ is the parameter value of position $s \in S$ simulated with model setup $j \in [1, J]$ at discrete time $t \in [t_1, t_T]$. Note that the complete set of $\left\{Q_j^s(t)\right\}_{j \in [1,J]}$ considers all possible combinations of implementations of all model arguments.* "

- The overview table (former Tab. A1, now Tab. 1) has been updated accordingly and is now part of the definition subsection as suggested by Reviewer 1 (Sect. 2.1, new count).

- Compressed the formulation of covariances and mean from combined sensitivities (old Eq. 3 and 4) as suggested by Reviewer 2. The highly similar equations 3 and 4 are now compressed to one equation (Eq. 9).

- In accordance to the rearrangements and extended description in new Sect. 2.1, the following sentence was included to explain the scope of sensitivities (new Sect. 2.2, l.128-129, new count):

  " *Since the definition of sensitivities refers to all possible combinations of implementations of all model arguments, the set of $\left\{X_j^s\right\}_{j \in [1,J]}$ is also denoted as set of combined sensitivities.* "

- A more detailed comment on the limitations of the independent assumption for atmospheric model parameters (l.489-497, new count, as suggested by Reviewer 2):

*" The results presented in this study demonstrate a considerable reduction of required computational resources under the assumption of independent sensitivities. As this method assumes linearity of parameter sensitivities, it may not be a sufficient approximation for all atmospheric parameters. However, the linear assumption relates to sensitivities of the perturbed parameters to model configurations. In other words, nonlinear effects resulting from the combination of different model arguments are disregarded. Note that this does not affect the impact of the parameters on prognostic fields like trace gas distributions and their propagation in time, which is beyond the ensemble generation and may still be highly nonlinear. Thus, the presented approach may also be suitable to be applied to parameters in NWP with highly nonlinear model dynamics. The sufficiency of this approximation for a model parameter in relation to the reduction of computational efforts needs to be evaluated for each specific application setup. "*

- Visualization of number of sensitivities required for combined and independent sensitivities as suggested by Reviewer 1.

**The applied terms of "model argument", "model parameter", "model input" and "model implementation" are confusing. For example, I think that "model configuration" is a better term for what is meant by "model argument".**

Reply4: This point was also considered in the modification of the description. We adopted the table (old Table A1 in Apx. A) and moved it to the related section in order to provide a better overview about the terms and variables including examples. We hope that this helps the reader to understand the mathematical description.

**Section 3 requires more clarity on what its objective and scientific content is. For example, it should be made much clearer that one of its main purposes is to compare the combined and the independent sensitivities approach. Further, concrete evidence of the successfulness of the method (see above) should be much more the focus of that section. Finally , it would be beneficial to provide a more science-based discussion of the soundness of the derived isoprene emissions ensemble.**

Reply5: In addition to the evaluation included in Sect. 3 (old count) as described in Reply1, we did the following modifications to the section in order to make its objective and structure clearer:

- A new introductory paragraph was added which describes the general objective of the section as well as the specific objectives of the its subsections (Sect. 4, l.296-303, new count):

*" This section provides results of the KL ensemble generation algorithm for an application to biogenic emissions in a regional CTM system. The modeling system used for the calculation of sensitivities and the specific setup of the algorithm are described in Sect. 4.1 and Sect. 4.2, respectively. Based on these, the results are presented with respect to two main objectives: Firstly, the behavior of the algorithm is illustrated for two different setups using combined and independent sensitivities, respectively. Sect. 4.3-4.5 present the results for each of the three major steps of the algorithm as described in Sect. 3. The description focuses on similarities between the two methods rather than on the detailed description of specific patterns. For the setup of independent sensitivities, additional uncertainties are included as described in Sect. 4.2 to demonstrate the inclusion of those. Secondly, the performance of the algorithm is evaluated for the two different setups. A comprehensive a-posteriori evaluation would ideally be based on a representative amount of data covering multiple conditions. However, observations of biogenic gases are rare and do only provide information on local concentrations, not on their emissions itself. As concentrations are affected by other uncertain processes, an ensemble of emissions or emission factors produced by the algorithm cannot be evaluated by observations alone. Therefore, Sect. 4.6 evaluates the performance of the algorithm in terms of ensemble statistics. "*

- The existing description of the modeling system and setup used in the section has been entitled by two separate subsections "Modeling System" and "Setup", respectively, in order to improve the readers' orientation (Sect. 4.1 and 4.2, new count).

- Two paragraphs describing the specific setup of combined and independent sensitivities are moved from Sect. 4.3 (new count) "sensitivity estimation" to the new subsection "setup" (now Sect. 4.2, l.340-346, new count). The content of the paragraphs itself were not changed.

- The scientific soundness of the generated emission ensemble is related to two aspects: Firstly, the soundness of the ensemble to represent leading uncertainties given by the sensitivities. This aspect is has been addressed by the new Sect. 4.6 as described above (see reply1). Secondly, the scientific meaning of the sensitivities. Surface properties (including land use information, drought response of plants and the land surface model) appear to induce the largest uncertainties to biogenic emissions of all considered trace gases. This aspect has been addressed in detail by a separate study (Vogel and Elbern, 2021a, DOI: acp-21-4039-2021). The following sentences have been added in the manuscript (l.413-415, new count):

  *" Thus, the comprised KL ensemble is restricted to dominant uncertainties indicated by the leading eigenmodes. In this case, these uncertainties are mainly induced by sensitivities to various surface conditions, which is in accordance to Vogel and Elbern (2021a). "*

**The discussion and conclusion section is very long. I would strongly recommend to introduce a separate and more concise conclusion section.**

Reply6: We thank the reviewer for this suggestion. We agree that it is worth dividing the concluding section into a separate discussion (Sect. 5, new count), followed by a shorter conclusion (Sect. 6, new count). This becomes even more convenient as some new aspects are included into the discussion following the remarks of Reviewer 1 and Reviewer 2.

**Figure captions need to provide more detail to make sure the figures can be understood without a lot of additional information. Acronyms in the figures should be spelled out.**

Reply7: We agree that especially the figures showing the sensitivities need more information and extended the captions accordingly. Additionally we added numbers to all subplots in Fig. 3 (new count) to make it more understandable.

- The description in caption Fig. 3 (new count) reads now:

  *" Set of combined sensitivities of isoprene emissions. Shown are isoprene emission factors for different combinations of model arguments as simulated by EURAD-IM. Emission factors are temporally averaged ratios of emissions divided by the reference emissions. The specific setup used for each of the 32 sensitivities given in Tab. C1. The sensitivities are divided into those using USGS (a) and MODIS (b) land use information, numbers attached to the individual subplots refer to the numbers in Tab. C1. In addition, a short abbreviation of the setup is given above each subplot. See also Vogel and Elbern (2021a) for a detailed description of the abbreviations and model implementations. Some major cities (Verona, Bologna, Modena) are indicated by their initial letters. "*

- Caption of Fig. 4 (new count):

  *" Set of independent sensitivities of isoprene emissions simulated by EURAD-IM. Shown are isoprene emission factors of simulations were only one model arguments differs from the reference setup. Plotting conventions including abbreviations as in Fig. 2. Additional uncertainties refer to the drought response (âno SMOISâ) and the emission model (âa-prioriâ) of biogenic emissions. The lower right subplot shows the independent mean from these sensitivities using Eq. (9a). A detailed description of the abbreviations and model implementations provided by Vogel and Elbern (2021a). "*

- Caption of Fig. 6 (new count) (the description of Fig. 8 refers also to this caption):

  " *Leading eigenvectors for combined sensitivities. The normalized eigenvectors are visualized as column of fields of different biogenic gases. Numbers above each column show the eigenvalues of the corresponding eigenvector.* "

- Caption of Tab. B1 (new count):

  " *Overview over of model setups used as combined sensitivities. A detailed description of the abbreviations and model implementations provided by Vogel and Elbern (2021a). The reference setup is denoted as \*, and deviations from this reference are written in bold letters ("PX" = Pleim-Xiu surface layer parameterization, "Du" = Dudhia shortwave radiation parameterization).* "

**Specific remarks:**

- **l3 replace "by" with "with"**

  Reply: The sentence was corrected accordingly (l.3, new count).

- **l5 make clearer what is novel about this approach (w.r.t application and method)**

  Reply: The description of the approach was modified to point out the novelty of this approach. The related part in the abstract now reads (l.4-7, new count):

  " *This study presents a novel approach, which substitutes the problem into a low-dimensional subspace spanned by the leading uncertainties. It is based on the idea that the forecast model acts as a dynamical system inducing multivariate correlations of model uncertainties. This enables an efficient perturbation of high-dimensional model parameters according to their leading coupled uncertainties.* "

- **l12 High spatial correlation of simulated biogenic emissions is not a surprise**

  Reply: Indeed, this is not the main point we wanted to make. We intended to point out the ability to represent highly correlated uncertainties by a few compounds. We hope that we could make this more clear by reformulating the sentence in the abstract as follows (l.13-14, new count):

  " *Rapidly decreasing eigenvalues state that highly correlated uncertainties of regional biogenic emissions can be represented by a low number of dominant components.* "

- **L 14 please compare 10 to the "uncondensed" number or ensemble member.**

  Reply: The simplest approach would be a random perturbation of each parameter at each location independently. Thus, the most "uncondensed" number of ensemble members equals at least the dimension of the parameters, which is $2 \cdot 10^6$ in this case study. This information was added to the referencing sentence in the abstract. Additionally, it might be important to note that the size of the KL ensemble can be readily adjusted to the required level of detail of the uncertainties (eg. if only the largest uncertainties need to be captured or also smaller contributions, compare reply to L 65 below). This aspect was also included in the abstract as well as the discussion section (l.470-474, new count, compare reply to L 361). The referencing sentence in the abstract reads now (l.14-15, new count):

  " *Depending on the required level of detail, leading parameter uncertainties with dimension of $\mathcal{O}(10^6)$ can be covered by a low number of about 10 ensemble members.* "

- **L 15 please provide more evidence that the 10-member ensemble indeed captures the uncertainty.**

Reply: This is closely related to the reviewers general remark, which was considered in Reply1. An evaluation was included in the manuscript (new Sect. 4.6, l.416-451, new count), but no additional information has been put in the abstract in order to keep it short.

- **L 20-40: Most of the discussions here is on NWP ensemble construction. The aim for NWP is to get realistic error growth with increasing lead time. That is not the primary objective for the CTM applications. Perhaps you could reference here to Reduced Rank Kalman Filter approaches for CTM.**

  Reply: The Reduced Rank KF approaches aim for dimension reduction for data assimilation, not uncertainty estimation. However, we see the similarities to our approach which is worth to mention. We added to following sentence in the introduction (l.47-49, new count):

  *" In the context of atmospheric data assimilation, reduced rank square-root Kalman filter approaches (Cohn and Todling, 1996; Verlaan and Heemink, 1996) have been successfully applied to reduce the high-dimensional covariance matrix to a small number leading eigenmodes (e.g., Auger and Tangborn, 2004; Hanea et al., 2004; Hanea and Velders, 2007). "*

- **L 44 it is not just the larger state vector which distinguishes NWP from CTM ensembles but the error growth characteristics.**

  Reply: We thank the reviewer for the suggestion to add this point. We included the following sentence at the referring part of the introduction (l.49-51, new count):

  *" Additionally, the temporal evolution of atmospheric chemical forecast errors differs from typical error growth characteristics in NWP. This inhibits a straightforward application of existing ensemble generation approaches from NWP to CTMs. "*

- **L 65 Please clarify what differentiates KL from singular vector decomposition methods.**

  Reply: Using the KL expansion for ensemble generation instead of singular vector approaches has one important advantage. In atmospheric modeling, singular vector decomposition (SVD) is mostly taken for complexity reduction which includes the time integration, where initial conditions are perturbed. In SV based ensemble generation approaches, each perturbation is generated by one singular vector scaled by its singular value. In case of linear conditions, this means that a specific number of perturbations is required depending on the number of considered singular vectors. In contrast, the KL expansion allows for a more flexible selection of the number of perturbations. Here, each perturbation $Y_{\omega_p}^s$ is sampled from the series of eigenmodes using different random numbers $y_d(\omega_p)$ for each perturbations (compare Eq.(15) ). Thus, the number of perturbations may be selected with respect to the desired level of detail and the available computational resources. Independent of the number of perturbations, the KL expansions ensures an optimal estimation of the largest uncertainties by the calculated perturbations.
  We thank the reviewer for noting this aspect, which we are happy to add in the manuscript. However the description requires knowledge about the KL expansion which is introduced in Sect. 3.3 (new count). Therefore, the description of differences from SVD methods was included in this section (Sect. 3.3, l.285-290, new count):

  *" Using the KL expansion for ensemble generation instead of singular vectors has one important advantage. In SV based ensemble generation approaches, each perturbation is generated by one singular vector scaled by its singular value. Using the KL expansion, each perturbation is sampled from the series of eigenmodes using different random numbers for each perturbations. This allows for a flexible selection of the number of perturbations depending on the desired level of detail. Independent of the number of perturbations, the KL expansions ensures an optimal estimation of the largest uncertainties by the calculated perturbations. "*

- **L 78 Please clarify in more detail the terms "model parameters" and "model arguments" and "model inputs"**

  Reply: This point has been addressed by Reply4. The manuscript has been modified accordingly.

- **L 85 The choice of the "differently configured model simulations" is in my opinion the most interesting aspect of the paper. Please expand if this choice is guided by the algorithm or by the (human) users.**

  Reply: We added this information as requested by the reviewer (l.183-185, new count):
  *" The configurations of the model simulations is selected by the user according to sources of uncertainties of the selected model parameters. For recurring applications of the ensemble generation algorithm, the selection may also be guided by results of previous applications of the algorithm. "*

- **L 81 "model inputs and configurations" that are very vague terms. Please provide more detail**

  Reply: This point was considered in the modification of the description of the algorithm as described in Reply4. Specifically, the examples of the terms have been included as follows (l.187-189, new count):
  *" Generally, atmospheric models are sensitive to their specific simulation setup including a large variety of model inputs and configurations like initial and boundary conditions, external input data and the selection of parameterization schemes in the model. "*

- **L 93 is it better to say the leading "un-coupled" ?**

  Reply: By "coupled" we meant the coupling between different parameters and locations. As this might be misleading, we decided to delete this word which is not of particular importance at this point. The phrase reads now as follows (l.194-195, new count):
  *" . . . , the extraction of leading uncertainties using a highly-parallelized eigenmode decomposition software (Sect. 3.2), . . . "*

- **L 98 please explain here how C is related to ensemble generation**

  Reply: Setting up the covariance matrix C is the first step towards ensemble generation. This remark was considered in the rearrangement of the description of the algorithm. We hope, that the connection is made clearer in the new version of the manuscript. For example in the introduction of the algorithm-section (new Sect. 3, l.193-194, new count):
  *" The algorithm consists of three major steps which are described in the following: the construction of parameter covariances from combined or independent sensitivities (Sect. 3.1) [. . . ] "*

- **L 99 clarify "model parameters" (all model grid points, a set of 20 land use classes ? )**

  Reply: We see that we did not sufficiently communicate this essential information in the manuscript. The term "model parameters" refers to any parameter in the prognostic equations of the model state, thus affecting the prognostic variables of the model. A description was included in the new Subsection 2.1 as described above (compare Reply3). Additionally, this was clarified at in the introduction of Sect. 3 describing the algorithm (l.178-179, new count):
  *" Here, a model parameter may be any parameter in the prognostic equations of the model state variables (compare Sect. 2.1). "*

- **L 108 - is model parameter a state vector element?**

  Reply: This is connected to the point above (remark L 99). We hope to made it clear by the above mentioned explanation.

- **L 112 It is not quite clear, why any model parameters can be considered stochastic**

  Reply: We intend to provide a slightly different view on the formulation of model parameters, but we see that this formulation might be confusing. Thus we decided to delete this part.

- **L 114 "i" was defined as model argument. parameters are Q (?)**

  Reply: Indeed, this is an error on our side, we meant "model argument" in this context. But this sentence was deleted according to the reply above.

- **L 125 is this triviality of importance later?**

  Reply: We wanted to note this point to assist the reader later in the description of the results. But as this point is also noted then, we agree that it can be deleted here. Therefore, we deleted the referring sentence (above l.128, new count).

- **L 126 the log distribution may need a bit more motivation. It may be justified for positive-definite variables such as concentration but not for all possible parameters.**

  Reply: The reviewer is right that not all parameters can be assumed to be lognormally distributed. We made this assumption for the specific application to trace gas emission for which it is pertinent. The sentences were modified as follows to make this more clear in the manuscript (l.128-131, new count):
  *" Depending on the type of model parameter, the sensitivity factors may not be Gaussian distributed. This is especially true for parameters which are positive by definition, like emissions of trace gases. Analogous to emission factors in Elbern et al. (2007), sensitivity factors of emissions are assumed to be lognormally distributed. In this case, the sensitivity factors are substituted to normally distributed sensitivities . . . "*

- **L 146 I do not understand the sub-sample argument here. Is not the same as choosing fewer arguments and implementations? If this is part is not referred to further down, I would simply omit it for the sake of clarity.**

  Reply: This argument was included in the manuscript in order to describe disadvantages of other approaches to reduce the number of required simulations. "sub-sampling" is technically not exactly the same as reducing the number of arguments or implementations, but all tree approaches would result in considerable deficiencies in the covariance estimation. As this argumentation appears to be more confusing than helping, we decided to remove it from the manuscript as noted in Reply3.
  Consequently, the "full combined" setup is now only denoted as "combined", the subscript "cf" used for specific quantities of the (full) combined setup is now replaced by "combi", e.g.: $J_{combi}$, $\mu_{combi}$, $C_{combi}$ and in Tab. 2.

- **L 159 The assumption of linearity is an important one. It contradicts the notion that sensitivities are non-linear. This mathematical choice need to be discussed for its realism for any specific application and choice of model arguments.**

  Reply: The reviewer is right. The need for a case-specific evaluation of this assumption was discussed in the concluding section. An additional note is made in the section containing the description of the algorithm, to which the reviewer is referring to (l.223-225, new count,

compare remark on L 187 below). A discussion on this in Sect. 5 was added as described above in Reply2.

- **L 180 I am not sure, if I understand all the need for the mathematical derivations here. Is not obvious that the assumption of linearity makes it not necessary to investigate the non-linear combined sensitivities. It seems not a result but an direct choice of assuming linearity?**

  Reply: Indeed, the assumption of linearity intend to avoid the need for combined sensitivities. Thus, the result is not surprising and we do not aim to provide a mathematical derivation on this. But for the comparison of the two methods (combined and independent sensitivities) it is important to compare the required number of simulations explicitly. As this is in accordance to one of the major remarks of Reviewer 1, we decided to keep this information in the manuscript and adopt it accordingly.

- **L 187 The assumption of tangent-linearity might also be a strong limitation for many atmospheric applications**

  Reply: We totally agree with the reviewer, the sentence he is referring to was intended to state this. We reformulated this sentence in order to state this more clearly (l.223-225, new count):
  *" While the equations are exact under the given assumption of tangent-linearity, this assumption might be a strong limitation for many atmospheric processes. "*

  Additionally, we added the following statement in the formulation of independent sensitivities which refers the reader to a more detailed discussion (new Sect. 2.3, l.174-176, new count):
  *" While the number of required simulations is considerably reduced, the underlying assumption of independent sensitivities disregards nonlinear interactions between different model arguments. A discussion of this assumption for the application to atmospheric model parameters is given in Sect. 5. "*

- **L 220 Please clarify what make KL superior to more standard methods to generate ensembles using singular vectors.**

  Reply: This aspect has already been included later in this subsection requested above (see reply to L 65). The mathematical optimality of the KL approach - which motivates its use in this algorithm - was additionally noted at the referring location (Sect. 3.3, l.258-259, new count):
  *" The KL expansion provides a mathematically optimal combination of the dominant directions of parameter uncertainties given by the leading eigenmodes. "*

- **L 240 Was the exp transformation prone to producing unrealistically high sensitivity factors?**

  Reply: This is the re-substitution for lognormally distributed parameters, as given in Eq. (2) (new count) of the manuscript, the description of which we adopted according to the reviewers remark on L 126 above. We agree that we have to make this more clear and also adopt the description analogous to the substitution above (l.280-281, new count):
  *" Finally, the ensemble of perturbations is transferred back to a set of perturbation factors $\left\{F^s_{\omega_p}\right\}_{s \in S}$. If the model parameters are assumed to be lognormally distributed, a resubstitution as counterpart of the logarithmic substitution in Eq. (2) is performed: [...] "*

- **L 255 A bit more detail on the generalization of the emission factors to model uncertainty parameter would be interesting.**

  Reply: The formulation of emission factors as uncertain model parameter in the KL algorithm was described later in the section in more detail. We see that the reader might already expect the information at this point, but this paragraph aims to provide more general information of

the system. So we added a note about this to the sentence as follows (l.315-316, new count):
*" The concept of emission factors used for emission rate optimization in EURAD-IM was adapted in the KL ensemble generation approach as described below. "*

- **L 255 "KL ensemble approach" I think the method is used to generate a ensemble. It is not yet an ensemble approach, which would mean the model simulation using the ensemble (emissions) itself. Please clarify what you mean.**

  Reply: The reviewer is right, it is an "ensemble generation approach" rather than an "ensemble approach" as we wrote mistakenly. We adopted the formulation everytime were it did appear in the manuscript.

- **L 265 Not clear what you mean "evaluation" ?**

  Reply: We agree that this is not the best word in this context. It was replaced by "description" (l.326, new count):
  *" The following description of the results focuses on emissions of isoprene, . . . "*

- **L 282 Please explain here that the reference is the default EURAD configuration and (2) the other configuration option.**

  Reply: This information were added to the manuscript as requested (l.340-341, new count):
  *" Two different implementations are selected for each argument ($R_i = 2$, $\forall\, i \in [1,6]$), were the reference $r_{i*} = 1$, $\forall\, i$ is the default configuration of EURAD-IM and $r_i = 2$ are the alternative implementations of each argument $i$. "*

- **L 285 Please discuss the plausibility of the shown sensitivity factors.**

  Reply: The sources of uncertainties leading to the sensitivity factors used here are discussed in detail in Vogel and Elbern (2021a). A statement on this has been added in the manuscript accordingly (l.344-346, new count):
  *" A detailed discussion on the origin of sensitivity factors used in this study is given in Vogel and Elbern (2021a). The selection of 32 combined configurations is based on the importance of the source of uncertainties reported there. "*

- **L 288 It is not clear what you mean by panel 1, 2**

  Reply: By the term "panel" we meant the upper left subplot of this Figure. The term was replaced by "subplot" at the position the reviewer is referring to as well as at other positions were it was used (l.360, 366, caption Fig. 3, new count).

- **L 297 The additional MEGAN uncertainty might useful for an application but for the paper it may complicates the comparison of the combined and independent sensitivities**

  Reply: We agree with the reviewer in this point. This was considered in the rearrangement of the section (see modifications related to Reply1) in which the evaluation of the two methods (combined and independent sensitivities) is done with a reduced setup without the MEGAN uncertainty. In our view, this uncertainty is nevertheless important for two reasons. Firstly, - as the reviewer stated - it is important to consider this uncertainty from the application-based point of view. Secondly, it provides an example how additional uncertainties can be included in the ensemble generation approach.

- **L 324 Figure 7 should be referenced after Fig 6**

  Reply: That is a mistake on our side. The existing description of Fig 6 and 7 (old count) was rearranged in a way that it is first referred to Fig 7 (new count, compare l.385-392).

- **L 339 The sentence is not clear. Does it mean the resulting ensemble members differ a lot between independent and combined method. That is not an evidence that the reduction in sensitivity runs works, quit the opposite.**

  Reply: We agree that this sentence is confusing. In fact, it is the differences in the setups of the covariance matrix which limits the comparison of resulting perturbations. We reformulated the sentence accordingly (l.400-402, new count):

  *" The different setups of the covariances from combined and independent sensitivities prohibit a direct comparison of their perturbations. As the ensemble generation from leading eigenmodes does not differ between these two setups (compare Sect. 3.3), resulting perturbations are only shown for independent sensitivities. "*

- **L 361 It should be made clearer that the assumption of linearity is the main reason for reduction of the sampling space, which is no surprise.**

  Reply: To be specific, the sampling space is already reduced significantly by setup of combined sensitivities. This is shown by the fast decrease of eigenvalues in Fig. 5 (new count). We thank the reviewer for piking up this point as we see that this needs to be discussed more in detail in the manuscript. We added following paragraph to the newly organized discussion section (Sect. 5, l.463-474, new count):

  *" By extracting leading eigenmodes from parameter covariances, the problem is transferred to a low-dimensional subspace spanned by the set of leading eigenmodes. This makes the approach highly efficient in covering dominant uncertainties compared to random perturbations at each location. For this un-condensed random approach, perturbations would be sampled from the complete $N$-dimensional space given by all considered model parameters at all grid-boxes. Compared to the uncondensed approach, the sampling space from which the KL perturbations are sampled is reduced to a $D$-dimensional eigen-mode subspace. The higher the parameter correlations, the less eigenmodes need to be considered in order to obtain sufficient sampling of uncertainties. Most atmospheric chemical parameters have high spatial- and cross-parameter-correlations which enables $D \ll N$ and thus significant reduction of the sampling space. In the results presented in this study, this has been demonstrated for a set of biogenic emissions with dimension $2 \cdot 10^6$. In this case, that sampling about 10 ensemble members from the leading eigenmodes is sufficient to cover the leading uncertainties of the high-dimensional parameters. The required numbers of eigenmodes and ensemble perturbations depend on the desired level of detail of the ensemble. Independent of the level of detail, the KL expansions ensures an optimal estimation of covariances by the calculated perturbations. "*

- **L 366 Please clarify what you mean by "performance of the ensemble"**

  Reply: In this context, the "performance of the ensemble" refers to the question, how well the ensemble of perturbations represents the true uncertainties of the parameters. This was included in the manuscript as follows (l.456-457, new count):

  *" Consequently, the accuracy of the KL ensemble to represent the true uncertainties crucially relies on the formulation of the covariances. "*

- **L 370 Please clarify, if this has been demonstrated by the presented practical results, or if this is only an assumption.**

  Reply: This is not demonstrated by the presented results, but is assumed based on the formulation of the algorithm. We agree that this needs to be clarified and we adopted the sentence

as follows (l.460-462, new count):

*" Although the greatest benefit is achieved for highly correlated parameters, the algorithm is assumed to efficiently combine the major uncertainties even for uncorrelated parameters. In case of missing correlations and therefore lack of ensemble reduction potential, the KL approach retains more leading eigenmodes and does not suppress required degrees of freedom. "*

- **L 379 Please explain the context of the assumed number of considered arguments and realizations**

  Reply: The information on the number of considered arguments and realizations was added. The referring passage reads now as follows (l.527-529, new count):

  *" Considering 2 realizations of 6 model arguments, only 7 independent out of a total number of 64 combined sensitivities are required. Thus, the method of independent sensitivities reduces the computational effort of model simulations prior to ensemble generation tremendously. "*